# The fundamental limits of least-privilege learning

## Abstract

Offloading model training and inference to a service provider is a common practice but raises concerns about data misuse when the provider is untrusted. Collaborative learning and model partitioning aim to address this issue by having clients share a representation of their data instead of the raw data itself. To prevent unintended information leakage, the feature mappings that produce such representations should follow the least-privilege principle, i.e., output representations that are relevant for the intended task, and nothing else. In this work, we provide the *first formalisation of the least-privilege principle for machine learning*. We first observe that *every task comes with fundamental leakage*: at the very least, a representation shared for a particular task must reveal the information that can be inferred from the task label itself. Considering this, we formalise the least-privilege principle as a bound on the inference gain about the data behind the representation over what is already revealed through the task's fundamental leakage. We prove that, under realistic assumptions on the data distribution, there is an *inherent trade-off* between the utility of representations output by a feature mapping and the leakage of information beyond the intended task. In experiments on image classification, we confirm that any data representation that has good utility for a given prediction task also always leaks more information about the original data than the task label itself. We show that this implies that censoring techniques that hide specific data attributes cannot achieve the desired goal of least-privilege learning.

## 1 Introduction and Related Work

The need to reveal data to untrusted service providers to obtain value from machine learning as a service (MLaaS) puts individuals at risk of data misuse and harmful inferences. The service provider observes raw data records at training or inference time and might abuse them for purposes other than the intended learning task. For instance, an image shared with a provider for the purpose of face verification might be misused to infer an individual's race and lead to discrimination (Citron & Solove, 2022). Thus, the traditional MLaaS setup could be in conflict with the data protection principles of data minimisation and purpose limitation which demand that "data should only be collected for specified, explicit, and legitimate purposes and not further processed in a manner that is incompatible with those purposes" (European Parliament and Council of the European Union, 2016).

**Sharing Representations to Prevent Data Misuse.** Collaborative learning or model partitioning claim to prevent such misuse in model training and inference settings, respectively. In both cases, individuals share a feature representation of their raw data with the service provider, in the form of model updates in the collaborative-learning setting (McMahan & Ramage, 2017; Hao, 2019) and of feature encodings in the model-partitioning setting (Osia et al., 2018; Chi et al., 2018; Wang et al., 2018; Brown et al., 2022). Proponents of both techniques argue that, because individuals only share a representation of their data, and not the data itself, the service provider no longer has access to information that might be abused for purposes other than the intended task.

**Unintended Feature Leakage and Least-Privilege Learning.** Previous research shows that a passive adversary can infer data attributes that are unrelated to the learning task, or even reconstruct data records (Boenisch et al., 2023; Ganju et al., 2018; Melis et al., 2019; Song & Shmatikov, 2019). For instance, Song & Shmatikov (2019) show that features extracted from a gender classification model also reveal an individual's race. Even higher layer features, that are assumed to be more

learning-task specific, might lead to unexpected inferences (Mo et al., 2021). These examples show that limiting data access to feature representations does not necessarily prevent information leakage and thus does not fully mitigate the risk of data misuse associated with attributes other than the learning task.

Some works suggest that the solution to this issue is to train models under the *least-privilege principle*. That is, to enforce that the representations shared with the service provider, e.g., gradients at model training time, or feature activations at inference time, only contain information relevant to the learning task, and nothing else (Melis et al., 2019; Brown et al., 2022). The concept of such least-privilege learning, however, has been only described informally, and lacks a precise definition. As a result, it is unclear how to evaluate whether a given representation fulfils this principle.

**Contributions.** In this paper, we make the following contributions towards understanding the limits of the least-privilege principle in machine learning:

i. We provide the first formalisation of the least-privilege principle for machine learning as a variant of the generalized Conditional Entropy Bottleneck problem (Fischer, 2020).

ii. We observe that any predictive model always reveals information that can be inferred from the prediction task label itself. We show experimentally that such *fundamental leakage* can reveal information that is not intuitively related to the prediction task. This could be misused for harmful inferences in a breach of data subject's expectations.

iii. We formally prove a fundamental trade-off: under realistic assumptions on the data distribution (e.g., label noise), whenever the learned representations have any utility, we must allow unintended leakage beyond the fundamental one.

iv. We experimentally demonstrate this trade-off. We show that so long as the representations have utility, there exist attributes different from the intended task label that can be inferred from the representations, even if we apply attribute censoring techniques.

## 2 PROBLEM SETUP

In this section, we motivate the least-privilege learning problem and formalise our setup. For ease of presentation, we focus on the *model partitioning*, which aims to prevent unintended information sharing at the inference stage (Osia et al., 2018; Chi et al., 2018; Mo et al., 2021). However, we note that all of our formal results apply to any setting in which feature representations are used as a means to limit the data revealed to untrusted third parties, such as the collaborative learning setting where individuals share gradients in place of raw data records.

**Notation.** Let $X, Y, S \sim P_{X,Y,S}$ be a set of random variables distributed according to $P_{X,Y,S}$ where $X \in \mathbb{X}$, and $Y \in \mathbb{Y}$ are, respectively, an *example* and its *learning task label*, and $S \in \mathbb{S}$ is a *sensitive attribute*. In contrast to related works on attribute obfuscation, we do not assume $S$ to be fixed (Melis et al., 2019; Song & Shmatikov, 2019; Zhao et al., 2020; Brown et al., 2022). Instead, we suppose that inference of any data attribute, other than the learning task, might result in harm and must be prevented. For any three random variables $X, Y, W$, we denote by $Y - X - W$ a *Markov chain*, which is equivalent to stating that: $Y \perp\!\!\!\perp W \mid X$.

**Assumptions on the Data Distribution.** To make our formal analyses tractable, we assume that the spaces $\mathbb{X}, \mathbb{Y}, \mathbb{S}$ are discrete and finite; and that the data domain is non-trivial: $|\mathbb{X}| > 1$ and $|\mathbb{Y}| > 1$. We also assume that the space of inputs is a subset of the space of sensitive attributes, $\mathbb{X} \subset \mathbb{S}$. In the worst case the sensitive "attribute" $S$ could be the input in its entirety. We also make the following assumption about the data distribution:

**Assumption A** (Strictly positive posterior). *We say that the* posterior distribution*, $P_{Y|X}$, is strictly positive if for any $x \in \mathbb{X}, y \in \mathbb{Y}$ we have $P_{Y|X}(y \mid x) > 0$.*

This assumption is realistic in settings where there exists inherent uncertainty about the ground truth label of a given example. Examples include the presence of *label noise* introduced by the labelling process (Song et al., 2022), and, under the Bayesian interpretation of probability, task labels that are subjective. This is the case, for instance, in many prediction tasks where model partitioning is used, such as emotion recognition or prediction of face attributes (e.g., smiling) which come with inherent uncertainty and labelling unreliability (Raji et al., 2021).

**Model Partitioning.** In the model partitioning setting, individual users hold a set of unlabelled examples $x_i$ for $i = 1, \ldots, N$ and would like to obtain predictions on these examples for their learning task label $y_i$ while hiding the value of their sensitive attribute $s_i$. To do so, users want to make use of a prediction service run by an untrusted third party, the *service provider*. This provider uses a *predictive model* $f(X) = \hat{Y}$ which, given an example $X$, outputs a label $Y$. The key question is how users can share enough information about $X$ with the service provider to obtain a correct prediction on $Y$ but prevent inference of any $S$ that may result in harm to the user.

During the training stage, depicted in Fig. 1 (*left*), the service provider runs a supervised training algorithm $M(D_T) = f$ on a train set $D_T \sim P_{X,Y,S}^T$ which outputs a trained model $f$. $D_T$ is an i.i.d. sample of size $T$. We assume that a model $f = f_E \circ f_C$ can be decomposed into a *feature mapping* $f_E$, and a *classifier* $f_C$. The feature mapping $f_E(X) = Z$ maps example $X$ to a model-specific *representation* $Z \in \mathbb{Z}$. We say that the feature mapping $f_E$ is non-trivial if it is neither constant nor completely random. The classifier $f_C(Z) = \hat{Y}$ produces a prediction $\hat{Y}$ based on an example's representation $Z$.

The goal of the learning process is to find a feature mapping $f_E$ that maximises the utility of the representations $Z = f_E(X)$ for the classifier $f_C(Z) = \hat{Y}$. One way to formalise this objective is the mutual information between the learned representations and the label $I(Y; Z) = I(Y; f_E(X))$ (Alemi et al., 2016). The higher the mutual information between $Z$ and $Y$, the more useful the representation will be for the prediction task.

During the inference stage, users submit examples $x_i$ to the trained model $f$ for prediction. Under *model partitioning*, the model is split into a client-side feature mapping and a server-side classification part. For each example $x_i$, users locally extract its representation as $z_i = f_E(x_i)$, and then send this representation to the service provider. The service provider returns $f_C(z_i) = \hat{y}_i$.

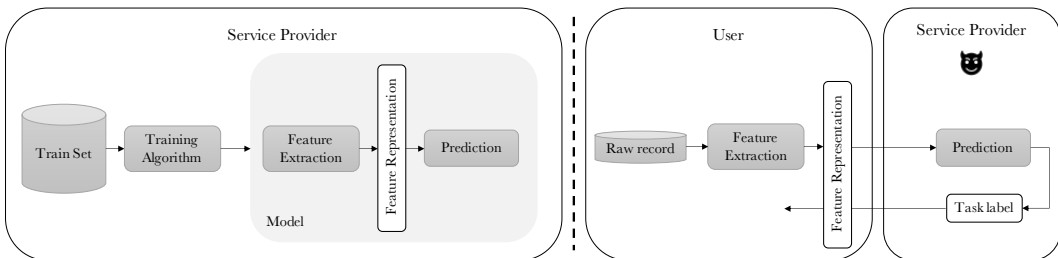

Figure 1: Overview over training (*left*) and inference (*right*) stages of model partitioning

The goal of model partitioning is to reduce the risk of data misuse and its potential harms. The assumption is that feature representations that are optimized to perform well on a specific task do not contain any information that could be re-purposed to predict information other than the task (Osia et al., 2018; Chi et al., 2018; Mo et al., 2021). It has been shown, however, that model partitioning is not enough to achieve this goal. Previous works thus suggest to learn representations under a "least-privilege principle" (Melis et al., 2019; Brown et al., 2022), but so far this idea has not been formalised. Our goal is to formalise this principle and characterize its feasibility.

## 3 THE LEAST-PRIVILEGE PRINCIPLE IN MACHINE LEARNING

In the following, we introduce the first formalisation of the least-privilege principle (LPP) for the machine learning domain. Due to space constraints, we defer all proofs to Appendix B.

**The Least-Privilege Principle.** The LPP is a design principle for building secure information systems introduced by Saltzer & Schroeder (1975). Its initial definition demands that "Every program and every user of the system should operate using the least set of privileges necessary to complete the job." In secure-systems engineering, a privilege is a clearly defined action that an actor in the system is authorised to carry out. Transferring this concept to the machine learning domain is not trivial. To do so, we have to first quantise the learning process into a set of smaller privileges and then define what is the minimum set of privileges needed to carry out a learning task.

**Attribute Inference.** We approach this problem through the lens of attribute inference: we assume that inference of any data attribute, other than the learning task, might result in harm. We hence define a privilege as the ability to learn the value of a particular attribute and formalise the LPP in terms of inference gain about data attributes other than the learning task.

In line with the standard practices in security and privacy, we analyse worst-case inference risks. To capture such worst-case risk of inferring $S$ from information $W$, we use Bayes-optimal adversaries that achieve optimal inference accuracy:

$$\hat{S}(W) \triangleq \arg \max_{g:\; \mathbb{W} \to \mathbb{S}} \Pr[S = g(W)]. \tag{1}$$

We use $\hat{S}$ without the argument to denote the baseline guess: $\hat{S} \triangleq \arg \max_{s \in \mathbb{S}} \Pr[S = s]$, representing the choice of the majority attribute value.

To measure the risk, we use *multiplicative gain* of an adversary $\hat{S}(W)$ who has access to information $W$ over the baseline guess:

$$I_\infty(S; W) \triangleq \log \frac{\Pr[S = \hat{S}(W)]}{\Pr[S = \hat{S}]}. \tag{2}$$

We also denote the gain of the adversary $\hat{S}(W, W')$ with access to two sources of information $W$ and $W'$ over the the guess with only one source of information $W'$ as:

$$I_\infty(S; W \mid W') \triangleq \log \frac{\Pr[S = \hat{S}(W, W')]}{\Pr[S = \hat{S}(W')]}. \tag{3}$$

In both cases, $\log(\cdot)$ is the base-2 logarithm. Such multiplicative gain has also been called multiplicative leakage (Braun et al., 2009), and shown (Liao et al., 2019) to be a special case of Arimoto's $\alpha$-mutual information (Arimoto, 1977) of order $\alpha = \infty$, hence the notation. Note that $I_\infty(S; W)$, unlike Shannon's information, is asymmetric.

### 3.1 STRAWMAN APPROACH: UNCONDITIONAL LEAST-PRIVILEGE PRINCIPLE

We start by formalising a strict interpretation of the LPP. In this interpretation, sharing the model-generated features instead of the raw data prevents *any leakage* for any attribute $S \neq Y$. Formally:

**Definition 1** (Unconditional LPP). *Given a data distribution $P_{X,Y}$, a feature map $Z = f_E(X)$ satisfies the unconditional LPP with parameter $\gamma$ if for any attribute $S \neq Y$ which follows the Markov chain $S - X - Z$, the attribute inference gain is bounded:*

$$\frac{\Pr[S = \hat{S}(Z)]}{\Pr[S = \hat{S}]} \leq 2^\gamma \tag{4}$$

*Equivalently:*

$$\sup_{S \neq Y:\; S - X - Z} I_\infty(S; Z) \leq \gamma \tag{5}$$

Previous work assumes that it is possible to find a feature map $f_E : \mathbb{X} \to \mathbb{Z}$ that fulfils the unconditional LPP, and at the same time produces representations with high utility for the learning task (Pittaluga et al., 2019; Brown et al., 2022). Next, we formally characterize this trade-off: *Can we achieve high utility, in the sense of $I(Y; Z)$, while simultaneously satisfying unconditional LPP?*

**Theorem 1** (Unconditional LPP and Utility Trade-Off). Suppose that $P_{Y|X}$ is strictly positive (Assumption A). Then, the following two properties cannot hold at the same time:

(1) $Z = f_E(X)$ satisfies the unconditional LPP with parameter $\gamma$     (2) $I(Y; Z) > \gamma$     (6)

We provide a full proof in Appendix B. This result implies that, whenever a representation has a certain utility for the learning task with $I(Y, Z) > \gamma$, there exists a sensitive attribute for which an adversary's inference gain is at least as large $I_\infty(S, Z) \geq \gamma$. In fact, it is easy to construct this attribute to be infinitesemally close to the task label $Y$ but not quite match it. In the next section, we provide an alternative, less literal, formalisation of the LPP that captures the requirement $S \neq Y$ yet precludes such cases.

### 3.2 Formalisation of the Least-Privilege Principle

We observe that, although it is impossible to hide *all* information about $X$, it is also an unnecessarily restrictive goal. To use the prediction service, users must be willing to reveal to the service provider at least the intended result of the computation $Y = f(X)$. As a consequence, they cannot conceal from the service provider *any information that can be inferred from $Y$ itself*. This information hence defines the *least privilege* that can be given to the service provider, i.e., the minimum access to data attributes that must be granted to carry out a task. We call this information the *fundamental leakage* of the task. For a given attribute $S$, the fundamental leakage equals $\Pr[S = \hat{S}(Y)]$.

We propose a formalisation of the LPP that only demands that sharing a record's feature representation $Z = f_E(X)$ does not reveal *more information* about a sensitive attribute $S$ than publishing $Y$ itself:

**Definition 2** (LPP). *Given a data distribution $P_{X,Y}$, a feature map $Z = f_E(X)$ satisfies the LPP with parameter $\gamma$ if for any attribute $S$ which follows the Markov chain $S - (X, Y) - Z$, the attribute inference gain from observing $(Z, Y)$ over the fundamental leakage is bounded:*

$$\frac{\Pr[S = \hat{S}(Z, Y)]}{\Pr[S = \hat{S}(Y)]} \leq 2^\gamma \tag{7}$$

*Equivalently:*

$$\sup_{S:\ S-(X,Y)-Z} I_\infty(S; Z \mid Y) \leq \gamma \tag{8}$$

Notably, the quantity constrained by the LPP is known as *maximal leakage* (Issa et al., 2019):

$$\mathcal{L}(X \to Z \mid Y) \triangleq \sup_{S:\ S-(X,Y)-Z} I_\infty(S; Z \mid Y). \tag{9}$$

In comparison to the unconditional variant, this formalisation does not require $S \neq Y$. Therefore, it does not restrict the adversary's absolute gain from observing $Z$, but only restricts the leakage about sensitive attribute $S$ *to its fundamental limit*, i.e., the leakage caused by the learning task label itself.

**Interpretation.** A feature map that satisfies the LPP in Definition 2 with a value of $\gamma \approx 0$ restricts the information available to the service provider to what is necessary for the intended purpose of the system: produce an accurate prediction of the task label $Y$. Hence, this formalisation supports the data protection principle of *purpose limitation* (see Section 1).

We must stress that, despite minimizing the information available to the service provider, this definition does not imply that a feature map that satisfies the LPP will necessarily prevent any harmful inferences. It only demands that access to $Z$ does not increase the risk over the risk already incurred by revealing $Y$. In Section 4.1, we show that even the fundamental leakage caused by correlations between the learning task and potentially sensitive data attributes, can violate users' expectations about what information they are willing to reveal, e.g., through a violation of contextual integrity Nissenbaum (2004), and lead to substantial harms Citron & Solove (2022).

**The Trade-Off.** We now study whether we can find a feature map that achieves the LPP in Definition 2, and simultaneously has good utility for the intended learning task in terms of mutual information $I(Z, Y)$. This question is a variant of the generalised conditional entropy bottleneck (CEB) problem (Fischer, 2020), which, in turn is a variant of the standard information bottleneck problem (Asoodeh & Calmon, 2020; Tishby et al., 2000). The original formulation of the CEB problem constrains the conditional mutual information between the input and the representation $I(X, Z \mid Y)$ rather than the maximal leakage $\mathcal{L}(X \to Z \mid Y)$. For our purpose, we require a leakage measure that captures the original claim from prior work that it is possible to learn representations that protect against *any* harmful inferences, i.e., protect even against *worst-case inference* adversaries. We hence consider maximal leakage. A different, but related, problem are Privacy Funnels (PFs); as Asoodeh & Calmon (2020) point out, these problems (PF and bottleneck-type problems like CEB) are "duals" of each other. PFs only consider *a single fixed sensitive attribute*. Here, to adequately model the least-privilege claim that a representation leaks *nothing* else other than the fundamental leakage, we have to treat any attribute other than the task as sensitive.

**Theorem 2** (LPP and Utility Trade-Off). Suppose that $P_{Y|X}$ is strictly positive (Assumption A). Then, the following two properties cannot hold at the same time:

$$\text{(1) } Z = f_E(X) \text{ satisfies the LPP with parameter } \gamma \qquad \text{(2) } I(Y; Z) > \gamma \tag{10}$$

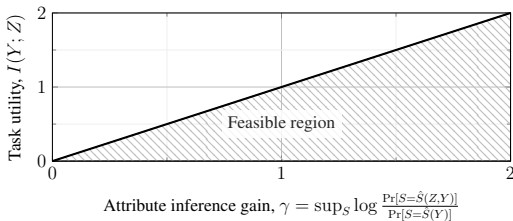

Figure 2: $\gamma$-LPP limits maximum utility $I(Y;Z)$ of a representation $Z$ to the greyed-out region.

See Fig. 2 for an illustration, and Appendix A for an interpretation of the trade-off in terms of task accuracy instead of mutual information.

The trade-off only exists because the LPP restricts the adversary's gain in terms of maximal leakage. Under weaker notions of leakage, such as in the standard CEB problem, there is no known trade-off (Fischer, 2020). Weaker leakage measures are not adequate, however, to formalise and assess the LPP which demands that the representations $Z$ leak *nothing else* other than the fundamental leakage.

One way to see why the trade-off holds is that there exists a set of attributes that reveal maximum possible information about $X$ from $Z$ (see Appendix B for a formal description). Intuitively, we would expect that by allowing for fundamental leakage, i.e., by conditioning on $Y$, we might reduce maximal leakage in those cases where $Y$ is one of these maximally revealing attributes. However, as we show in Appendix B, strict positivity of the posterior distribution (Assumption A) prevents the task label $Y$ to be such a maximally revealing attribute and the least-privilege and utility trade-off thus remains the same as under unconditional LPP (Theorem 1).

An important implication of Theorem 2 is that perfect LPP with $\gamma = 0$, where the feature representations do not allow for any inference gain over fundamental leakage, is only possible when the representations are constant or completely random:

**Corollary 1.** *Under Assumption A, a feature map $Z = f_E(X)$ can achieve perfect LPP with $\gamma = 0$ if and only if it is trivial: $Z \perp\!\!\!\perp X$.*

Theorem 2 implies that in many realistic applications (see Assumption A) of least-privilege learning we have a stringent trade-off between the representation's utility and the LPP. Notably, this trade-off holds for *any* feature representation *regardless of the way the feature representations are obtained.*

## 4 EMPIRICAL EVALUATION

In this section, we empirically validate our theoretical results. Our experiments confirm that if the feature representation learned by a model are useful for the intended learning task, there always exists a sensitive attribute which an adversary can infer with high accuracy. We demonstrate that this is a *fundamental trade-off that applies to any feature representation regardless of the model architecture or the feature learning technique*. Due to space constraints, we only present results for an image dataset under two different learning techniques for a single model architecture. In Appendix C, we show additional results that confirm that our theoretical results indeed hold on a much wider range of datasets, models, and learning techniques.

**Data.** We use the LFWA+ image dataset which has multiple binary attribute labels for each image (Huang et al., 2008). The full dataset contains $13,143$ examples which we split in the following way: $20\%$ of records are given to the adversary as an auxiliary dataset $D_A$. The remaining $10,514$ records are split $80/20\%$ across a train $D_T$ and evaluation set $D_E$.

**Model.** We choose a simple Convolutional NN (CNN256) with three spatial convolution layers with $32$, $64$, and $128$ filters, kernel size set to $(3, 3)$, max pooling layers with pooling size set to 2, followed by two fully connected layers of size 256 and 2. We use ReLU as the activation function for all layers. CNN256 mimics the model architecture used by Melis et al. (2019), the first work to propose feature learning under a least-privilege principle. We use the network's last layer representation as feature map $Z f_E(X)$.

**Adversaries.** To evaluate leakage of a given attribute, we instantiate Bayes-optimal adversaries with access to the auxiliary set $D_A$ of labelled examples $\mathbf{r}_i = (x_i, y_i, s_i)$ for $i =$

$1, \ldots, A$. The label-only adversary $\hat{S}(Y)$ computes the relative frequency counts over $D_A$ to estimate $\tilde{\Pr}[S = s \mid Y = y] = \sum_{i=1}^{A} \mathbf{1}[s_i=s, y_i=y] / \sum_{i=1}^{A} \mathbf{1}[y_i=y]$ and outputs a guess according to $\hat{S}(Y = y) = \arg\max_s \tilde{\Pr}[S = s \mid Y = y]$. The features adversary $\hat{S}(Z, Y)$ is given black-box access to the trained model $f = f_E \circ f_C$. To collect a train set, the features adversary submits each example $X_i$ in $D_A$ to the model and receives back its model-generated feature representation $z_i = f_E(x_i)$. The adversary then trains a Random Forest (RF) classifier with 50 decision trees on the collected samples $(z_i, y_i, s_i)$ to estimate $\tilde{\Pr}[S = s \mid Z = z, Y = y]$. We opt for RFs as an attack model based on its superior performance over other classifiers we tested.

**Experimental Setup.** In each experiment, we select one out of 12 attributes from the LFWA+ dataset as the model's learning task $Y$ and a second attribute as the sensitive attribute $S$ targeted by the adversary. We select attributes for which we expect the distribution $P_{Y|X}$ to be strictly positive due to their subjective nature (see A). We repeat each experiment 5 times to capture randomness of our measurements for both the model and adversary, and show average results across all 5 repetitions. At the start of the experiment, we split the data into the three sets $D_T$, $D_E$, and $D_A$. We train the model CNN256 on the train set $D_T$ for the chosen learning task and then estimate its utility on the evaluation set $D_E$. Because $Y$ is a (binary) discrete variable and $Z$ a high-dimensional, continuous variable, directly estimating $I(Y; Z)$ is not feasible. We instead measure utility by estimating the multiplicative gain $\tilde{I}_{\infty}(Y; Z) = \log \tilde{\Pr}(Y=\hat{Y}(Z)) / \tilde{\Pr}(Y=\hat{Y})$, where $\hat{Y}(Z)$ denotes the model's prediction for a record's task label $Y$ and $\hat{Y}$ without the argument the majority class baseline guess. After model training and evaluation, we train both the label-only and features adversary on the auxiliary data $D_A$. For a given sensitive attribute $S$, we estimate the adversary's gain as $\tilde{I}_{\infty}(S, Z \mid Y) = \log \tilde{\Pr}[S=\hat{S}(Z,Y)] / \tilde{\Pr}[S=\hat{S}(Y)]$.

**Learning Techniques.** We implement two learning techniques: (1) standard ERM with SGD and (2) attribute censoring to learn representations that hide a given sensitive attribute. For censoring, we use the gradient reversal strategy (GRAD) introduced by Raff & Sylvester (2018) with a censoring parameter of 100. We choose GRAD because it provides stable performance (it effectively hides the chosen sensitive attribute without large drops in model performance (Zhao et al., 2020)) and, in comparison to other censoring techniques, can be applied to any model architecture.

## 4.1 THE POTENTIAL HARMS OF FUNDAMENTAL LEAKAGE

To evaluate the potential harms of fundamental leakage, we measure it across 12 prediction tasks in the LFWA+ dataset.

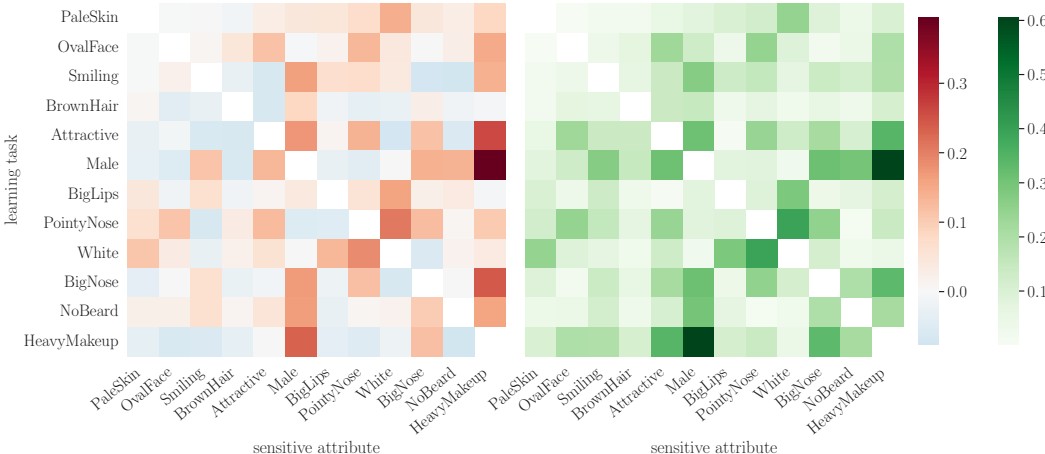

Figure 3: **Fundamental leakage: the task label reveals information about other data attributes; which might not be obvious to data subjects.** Attribute inference gain of the label-only adversary (*left*) and pairwise Pearson's correlation between attributes (*right*). In the LFWA+ dataset, the 'Attractive' label is highly correlated with the perceived gender. Thus, predicting the 'Attractive' label will reveal information about gender.

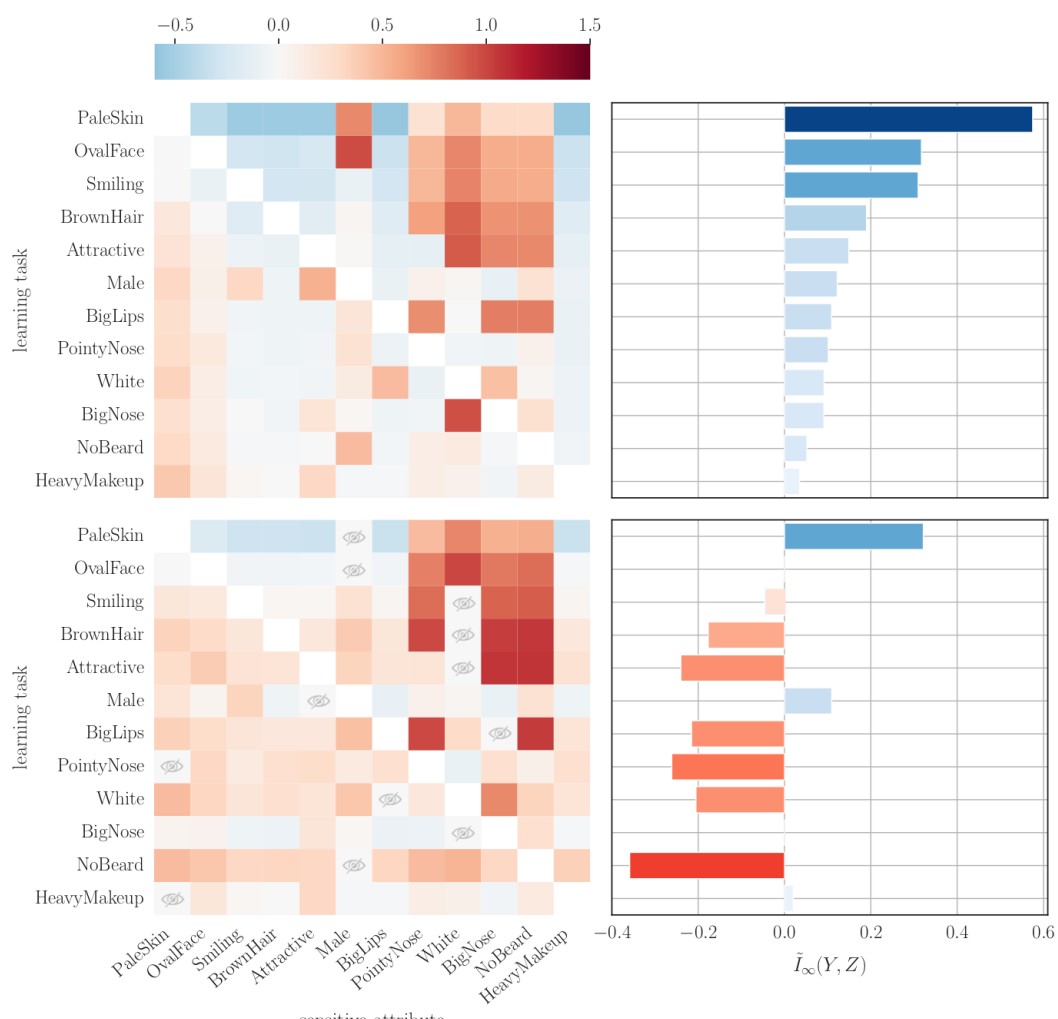

sensitive attribute

Figure 4: **If the model-generated representations have utility for the task (*right*), there exists a sensitive attribute with an even higher inference gain for the adversary (*left, red means more leakage*).** This holds for both standard ERM (*top*) and attribute censoring (*bottom*) where we censor the attribute with highest leakage in the respective ERM model (marked as 👁). Censoring has a 'whack-a-mole' effect: as we censor one attribute, leakage of another attribute increases.

Fig. 3 (*left*) shows the adversary's gain $\tilde{I}_\infty(S, Y) = \log \hat{\Pr}[S=\hat{S}(Y)]/\hat{\Pr}[S=\hat{s}]$ in predicting sensitive attribute $S$ from $Y$ for a combination of 12 learning tasks and sensitive attributes. It represents the fundamental leakage of a system that fulfils its intended purpose, i.e., that produces accurate class labels for the chosen learning task. The matrix in Fig. 3 (*right*) shows the absolute pairwise Pearson's correlation coefficient between attribute labels. The graph shows that, as expected, a strong linear relationship between the learning task and the sensitive attribute targeted by the adversary leads to a large fundamental leakage. For instance, 'Attractive' and 'Male' are mildly correlated negatively ($-0.3094$ correlation coefficient), but this already increases the adversary's gain in inferring sensitive attribute 'Male' when the learning task is 'Attractive'. Although the increase is small in this case, it illustrates how inferences due to a task's fundamental leakage can not only be counterintuitive, but also reveal information that could lead to harm (in this case, discrimination due to gender).

The fundamental leakage for some pairs of tasks and sensitive attributes is highly concerning. It implies that users of the prediction service reveal to the service provider not only their task label but also *any attribute that is correlated with the chosen task*. This consequence is rarely made explicit to users when they are informed about the data collection and processing, and might lead to unexpected harms beyond those associated with revealing the task label itself. In the example above, a user expecting to only reveal 'attractiveness' might not expect that their gender is revealed, with the

ensuing risks of discrimination. To better inform users about their risk, providers would have to list all sensitive attributes that might be leaked through a task's fundamental leakage. Knowing *all* such attributes is likely infeasible. To address this problem, service providers could empirically evaluate whether attributes considered particularly sensitive are part of the fundamental leakage and inform data subjects about the result.

## 4.2 THE LEAST-PRIVILEGE AND UTILITY TRADE-OFF

One way to interpret Theorem 2 is that whenever the features learned by a model are useful for a given prediction task, there always exists a sensitive attribute for which an adversary gains an advantage from observing a target record's feature representation. We experimentally show this fundamental limit of least-privilege learning.

Fig. 4 compares the trade-off between utility and attribute leakage of models trained with standard SGD (*top*) and with GRAD to censor the representation of a single sensitive attribute (*bottom*). The blue horizontal bars in Fig. 4 (*right*) show the model's utility for learning task $Y$ measured as $\tilde{I}_\infty(Y, Z)$. The heatmaps in Fig. 4 (*left*) show the difference between the adversary's inference gain and the model's utility $\Delta_{\mathsf{PUT}} \triangleq \tilde{I}_\infty(S, Z \mid Y) - \tilde{I}_\infty(Y, Z)$. Each row corresponds to a different learning task $Y$, each column represents a different sensitive attribute targeted by the adversary. In Fig. 4 (*top*), we see that in the LFWA+ dataset, for example, the features learned by a model that predicts attribute 'Smiling' increase the adversary's inference accuracy for attribute 'White'. Different tasks result in a high leakage for different attributes, e.g., 'OvalFace' reveals a lot of additional information about gender, while 'HeavyMakeup' is indicative of 'PaleSkin' and 'Attractive' but does not have much influence on the inference power of other attributes. Importantly, however, for *every* task, there is at least one sensitive attribute for which $\Delta_{\mathsf{PUT}} > 0$. These results confirm that the features learned by a model trained to perform well on its learning task reveal more information than a task's fundamental leakage, violating the least-privilege principle.

In Fig. 4 (*bottom*), we show results for models trained under attribute censoring, a common technique used to address unintended information leakage (Song & Shmatikov, 2019; Brown et al., 2022; Zhao et al., 2020). For each task, we censor the attribute with the highest leakage under standard training. First, as expected, censoring limits the leakage of the censored attribute, and decreases the utility of the model (Song & Shmatikov, 2019; Zhao et al., 2020). However, we observe the inherent trade-off from Theorem 2: In all our experiments, an adversary can find another data attribute for which $\Delta_{\mathsf{PUT}} > 0$ and that thus violates the least-privilege principle. In Appendix C, we show equivalent results for other learning techniques, such as, differentially private training and adversarial representation learning that aim to hide sensitive information about the original data $X$.

**Takeaways.** Our experiments provide evidence for *the fundamental trade-off* we derive in Theorem 2: the representations generated by models that perform well on their intended task fail to fulfil the LPP. Censoring techniques can be used to limit the adversary's inference gain on a particular attribute but cannot avoid the inherent trade-off. There always exists an attribute, other than the learning task, that violates the LPP. This 'whack-a-mole' effect is a phenomenon observed in related scenarios, such as, privacy-preserving data publishing (Narayanan & Shmatikov, 2019).

## 5 CONCLUSIONS

The promise of least-privilege learning – to learn feature representations that are relevant for a given task but avoid leakage of any information that might be misused and cause harm – is extremely appealing. In this paper, we show that in many realistic tasks where there exists inherent uncertainty about the task labels, *any* representation that provides utility for its intended task must always leak information about attributes other than the task. Even when the representations fulfil the least-privilege principle, it is not possible to provide technical guarantees that sharing those representations will not result in harms due a task's fundamental leakage. This is because the task predictions themselves are correlated with data attributes that are possibly sensitive. This issue applies to any settings in which users share representations instead of raw data records to limit data access, including model partitioning and collaborative learning. The issue cannot be addressed through censoring (Song & Shmatikov, 2019), which can only limit the inference of specific attributes, but cannot prevent leakage of *all* sensitive attributes.

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

Figure 5: Bound on the highest achievable task classification accuracy from maximum attribute inference accuracy. We provide the bound for illustrative purposes but, as it is loose, caution against using it to make decisions about acceptable leakage. In particular, for perfect LPP at the point $x = 0.5$ by Corollary 1, we must have trivial task accuracy 0.5, whereas this bound gives $\approx 0.8$.

## A    INTERPRET TRADE-OFFS THROUGH TASK ACCURACY

Our results on the fundamental trade-off in Theorem 2 relate maximum gain in attribute inference to mutual information $I(Y; Z)$. Next, we provide an interpretation of the trade-off in terms of task classification accuracy instead of mutual information. Specifically, we provide a trade-off between $\gamma$ and task accuracy when the representations are used by a classifier $f_C$ of a binary task label with uniform prior.

**Corollary 2.** Suppose that the following hold:

- We have binary classification ($\mathbb{Y} = \{0, 1\}$) with uniform prior ($P(y) = 1/2$).

- The features $f_E$ satisfy $\gamma$-LPR with such $\gamma$ that no binary attribute $S$ with uniform prior ($P(s) = 1/2$) and trivial fundamental leakage ($\Pr[S = \hat{S}(Y)] = 1/2$) can be inferred with accuracy higher than $\beta$ ($\Pr[S = \hat{S}(Z, Y)] \leq \beta$)

Then, no classifier $f_C$ can achieve the task accuracy higher than the following:

$$\max_{f_C:\, \mathbb{Z}\to\mathbb{Y}} \Pr[f_C \circ f_E(X) = Y] \leq 1 + \frac{\log(\beta)}{2\log\left(-6/\log(\beta)\right)}. \tag{11}$$

See Fig. 5 for an illustration. At $\beta = 1/2$ we have perfect LPP with $\gamma = 0$, thus by by Corollary 1 the task accuracy can be at most $1/2$. In Eq. (11), in this case the bound on task accuracy is only $\approx 0.8$, thus the bound is loose. This is because the convertion between mutual information and classification accuracy is loose.

*Proof.* First, observe that to achieve the condition on $\gamma$, we need to have:

$$\log \frac{\Pr[S = \hat{S}(Z, Y)]}{1/2} \leq \log(2\beta) = \gamma \tag{12}$$

By Theorem 2, we therefore have $I(Y; Z) \leq \log(2\beta)$.

From $I(Y; Z)$, we can obtain a bound on classification error of $Y$ from $Z$ by Fano's inequality (see, e.g. Cover & Thomas, 2006, Eqn.(2.140)), noting that $Y$ is binary and follows a uniform distribution, hence $H(Y) = \log(2)$:

$$\min_{f_C:\, \mathbb{Z}\to\mathbb{Y}} \Pr[f_C(Z) \neq Y] \geq H_2^{-1}(\log(2) - I(Y; Z)), \tag{13}$$

where $H_2(p) = -p\log(p) - (1-p)\log(1-p)$, is the binary entropy function. To have an easy-to-use expression, we use a bound due to Calabro (2009) (in (Zhao et al., 2020)):

$$H_2^{-1}(t) \geq \frac{t}{2\log(6/t)} \tag{14}$$

Thus, we have:

$$\max_{f_C:\, \mathbb{Z}\to\mathbb{Y}} \Pr[f_C(Z) = Y] \leq 1 - \frac{t}{2\log(6/t)}, \tag{15}$$

where $t = \log(2) - I(Y; Z) \geq \log(2) - \log(2\beta) = -\log(\beta)$. □

## B  FORMAL DETAILS

**Maximal Leakage.** First, let us provide some useful properties of maximal leakage.

Maximal leakage and conditional maximal leakage are defined (Issa et al., 2019) as follows:

$$\mathcal{L}(X \to Z) \triangleq \sup_{S:\ S-X-Z} I_\infty(S; Z) \quad \mathcal{L}(X \to Z \mid Y) \triangleq \sup_{S:\ S-(X,Y)-Z} I_\infty(S; Z \mid Y). \quad (16)$$

**Lemma 1** (Issa et al. (2019)). *Maximal leakage has the following properties:*

- *Maximal leakage bounds mutual information:* $\mathcal{L}(X \to Z) \geq I(X; Z)$

- *Maximal leakage and conditional maximal leakage have the following closed forms:*

$$\mathcal{L}(X \to Z) = \log \left( \sum_{z \in \mathbb{Z}} \max_{x \in \mathrm{supp}(X)} P(z \mid x) \right)$$

$$\mathcal{L}(X \to Z \mid Y) = \log \left( \max_{y \in \mathrm{supp}(Y)} \sum_{z \in \mathbb{Z}} \max_{x \in \mathrm{supp}(X \mid Y = y)} P(z \mid x, y) \right)$$

**Maximally Revealing Attributes.** We demonstrate that a certain family of attributes is maximally revealing, i.e., achieves the supremum $\sup_{S:\ S-X-Z} I_\infty(S; Z)$.

**Definition 3.** *A family of maximally revealing attributes $\mathcal{S}^*(X)$ is a subset of probability distributions over $\mathbb{S}$, defined as follows. Any attribute $S \in \mathcal{S}^*(X)$ is such that for any $x \in \mathbb{X}$ there exists a map $\mathcal{U}_S(x) : \mathbb{X} \to 2^\mathbb{S}$, and $S = s \in \mathcal{U}_S(x)$ if and only if $X = x$.*

**Lemma 2.** *The class of maximally revealing attributes leads to the highest inference gain:*

$$\sup_{S:\ S-X-Z} I_\infty(S; Z) = \sup_{S \in \mathcal{S}^*(X)} I_\infty(S; Z) \quad (17)$$

*Proof.* Denoting by $P_{\tilde{X}} \ll P_X$ the relation $\mathrm{supp}(P_{\tilde{X}}) \subseteq \mathrm{supp}(P_X)$, observe that any distribution $P_{\tilde{X}} \ll P_X$ can be represented as follows:

$$P_{\tilde{X}}(x) = \frac{\sum_{s \in \mathcal{U}_{S^*}(x)} P_{S^*}(s)}{\sum_{x \in \mathbb{X}} \sum_{s \in \mathcal{U}_{S^*}(x)} P_{S^*}(s)}, \quad (18)$$

for some $S^* \in \mathcal{S}^*(X)$.

Liao et al. (2019, Eq. 106–109) show that for $P_{\tilde{X}}$ and $S^* \in \mathcal{S}^*(X)$ defined as in Eq. (18), we have:

$$I_\infty(S^*; Z) = \sup_{S:\ S-\tilde{X}-Z} I_\infty(S; Z) \quad (19)$$

Moreover, Liao et al. (2019, Theorem 2) show that the following property holds:

$$\sup_{S:\ S-X-Z} I_\infty(S; Z) = \sup_{\tilde{X}:\ P_{\tilde{X}} \ll P_X} \sup_{S:\ S-\tilde{X}-Z} I_\infty(S; Z) \quad (20)$$

Using the fact that for any $\tilde{X}:\ P_{\tilde{X}} \ll P_X$ there exists $S^* \in \mathcal{S}^*(X)$ such that Eq. (18) holds, we combine Eq. (19) and Eq. (20) to get the sought result. $\square$

**Lemma 3.** *Suppose that $|\mathbb{X}| > 1$. If an attribute is maximally revealing, $S \in \mathcal{S}^*(X)$, then the conditional distribution $P_{S|X}$ is non-positive: there exist a $s \in \mathbb{S}$ and $x \in \mathbb{X}$ such that $P(s \mid x) = 0$.*

*Proof.* By contradiction. Suppose that for all $x \in \mathbb{X}$, we have $P(s \mid x) > 0$. But then, there exist $x' \neq x'' \in \mathbb{X}$ such that $s \in \mathcal{U}_S(x')$ and $s \in \mathcal{U}_S(x'')$, which contradicts the definition of $\mathcal{S}^*(X)$. $\square$

**Omitted Proofs.** Next, we provide the proofs of the formal statements in the main body of the paper.

*Proof of Theorem 1.* By construction, we have a Markov chain $Y - X - Z$. By data processing inequality, we thus have a bound on utility: $I(Y; Z) \leq I(X; Z)$. Next, observe that $Y \notin \mathcal{S}^*(X)$ by Lemma 3 and strict positivity (Assumption A). Therefore,

$$\sup_{S:\ S-X-Z,\ S \neq Y} I_\infty(S; Z) = \sup_{S \in \mathcal{S}^*(X)} I_\infty(S; Z) = \sup_{S:\ S-X-Z} I_\infty(S; Z) = \mathcal{L}(X \to Z) \quad (21)$$

by Lemma 2 and the definition of maximal leakage. Finally, applying a property of maximal leakage, we have $I(Y; Z) \leq I(X; Z) \leq \mathcal{L}(X \to Z) = \sup_{S:\ S-X-Z,\ S \neq Y} I_\infty(S; Z) \leq \gamma$. $\qquad\square$

**Remark 1.** *Theorem 1 only requires that $Y \notin \mathcal{S}^*(X)$. This holds under weaker assumptions than the strict positivity of the posterior (Assumption A). For instance, it is sufficient that there exist any $x, x', y$ such that both $P_{X|Y}(x \mid y) > 0$ and $P_{X|Y}(x' \mid y) > 0$.*

*Proof of Theorem 2.* First, we show that in our Markov chain and under the strictly positive posterior assumption, we have a surprising result that $\mathcal{L}(X \to Z \mid Y) = \mathcal{L}(X \to Z)$.

To see this, observe that maximal leakage has the following closed form by Lemma 1:

$$\mathcal{L}(X \to Z \mid Y) = \log \left( \max_{y \in \mathrm{supp}(Y)} \sum_{z \in \mathbb{Z}} \max_{x \in \mathrm{supp}(X|Y=y)} P(z \mid x, y) \right) \quad (22)$$

$$= \log \left( \max_{y \in \mathrm{supp}(Y)} \sum_{z \in \mathbb{Z}} \max_{x \in \mathrm{supp}(X|Y=y)} P(z \mid x) \right), \quad (23)$$

where the second equality is by the Markov chain $Y - X - Z$.

Next, observe that we have $P(x \mid y) \propto P(y \mid x) \cdot P(x)$ which, by assumption, is positive so long as $x \in \mathrm{supp}(X)$. As a consequence, the support of $X$ is independent of $Y$: $\mathrm{supp}(X \mid Y = y) = \mathrm{supp}(X)$, for any $y \in \mathbb{Y}$. Therefore, we can simplify the last form:

$$\mathcal{L}(X \to Z \mid Y) = \log \left( \max_{y \in \mathrm{supp}(Y)} \sum_{z \in \mathbb{Z}} \max_{x \in \mathrm{supp}(X|Y=y)} P(z \mid x) \right) \quad (24)$$

$$= \log \left( \sum_{z \in \mathbb{Z}} \max_{x \in \mathrm{supp}(X)} P(z \mid x) \right), \quad (25)$$

which is an equivalent form of $\mathcal{L}(X \to Z)$.

Finally, to obtain the trade-off, it suffices to observe that $I(Y; Z) \leq I(X; Z) \leq \mathcal{L}(X \to Z) \leq \gamma$, where the second inequality is by data processing in the Markov chain $Y - X - Z$, and the third is a property of maximal leakage: $\mathcal{L}(X \to Z) \geq I(X; Z)$. $\qquad\square$

One way to interpret this result is that the family of maximally revealing attributes $\mathcal{S}^*(X)$ is not sensitive to conditioning on $Y$ so long as $Y \notin \mathcal{S}^*(X)$, which we have by Lemma 3 and strict positivity (Assumption A).

## C  ADDITIONAL EXPERIMENTS

### C.1  VARYING MODEL ARCHITECTURE

To demonstrate that, as predicted by Theorem 2, the strict trade-off between features' utility for a downstream prediction task and the LPP applies regardless of a model's architecture or the structure of the feature encoder $Z = f_E(X)$, we conduct an additional experiment on the LFWA+ image dataset (see Section 4) using a different model architecture. We use the ResNet-18 architecture from He et al. (2015) implemented by PyTorch. Training batch size is 32, SGD learning rate is 0.01.

Fig. 6 compares the trade-off between utility and attribute leakage of a CNN256 (*top*) and a RESNET18 (*bottom*) models, both trained with standard SGD. The blue horizontal bars in Fig. 6 (*right*) show the model's utility for learning task $Y$ measured as $\tilde{I}_\infty(Y, Z)$. The heatmaps in Fig. 6 (*left*) show the difference between the adversary's inference gain and the model's utility

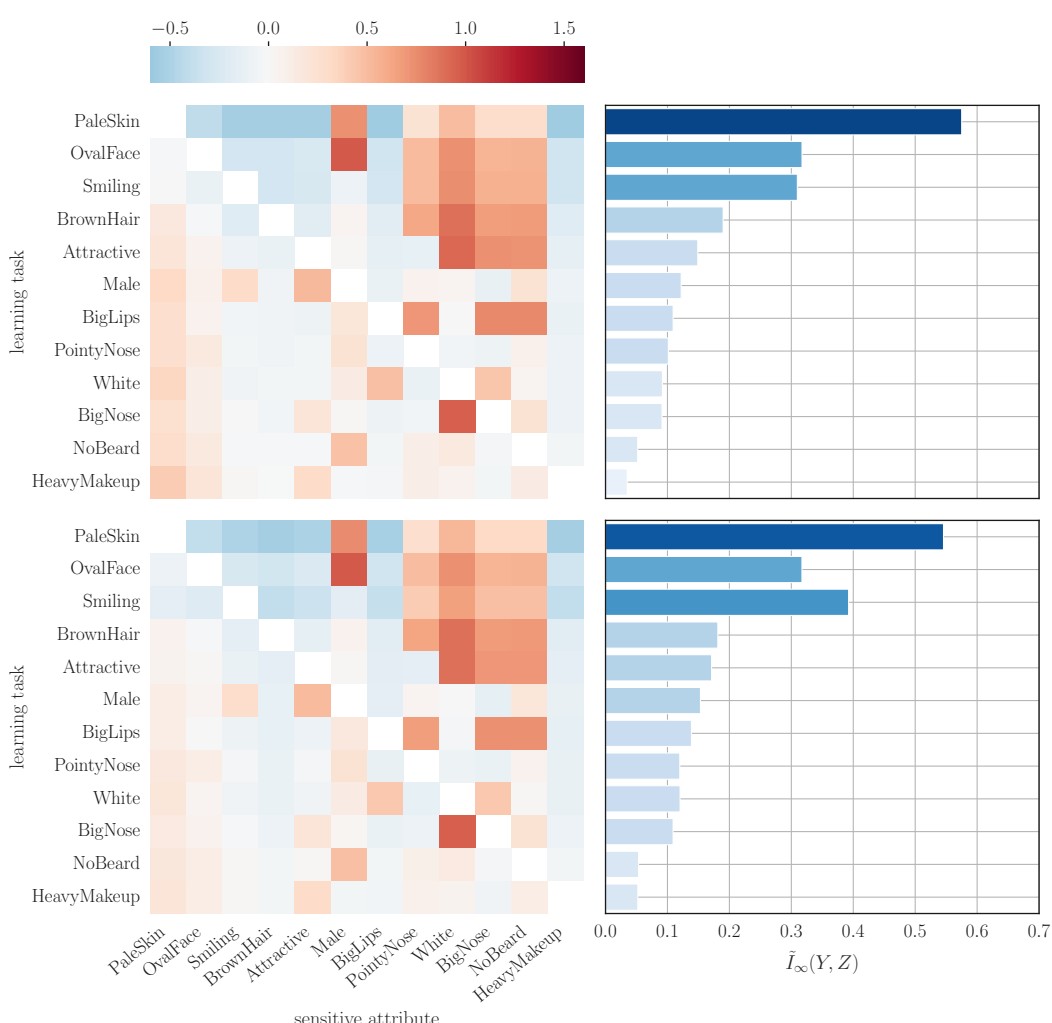

Figure 6: **If the model-generated representations have utility for the task (*right*), there exists a sensitive attribute with an even higher inference gain for the adversary (*left, red means more leakage*).** This holds for both a CNN256 (*top*) and a RESNET18 model (*bottom*), i.e., regardless of the model architecture.

$\tilde{I}_\infty(S, Z) - \tilde{I}_\infty(Y, Z)$. Each row corresponds to a different learning task $Y$, each column represents a different sensitive attribute targeted by the adversary. We observe that regardless of the model architecture, for any learning task there always exists a sensitive attribute for which $\tilde{I}_\infty(S, Z) > \tilde{I}_\infty(Y, Z)$ and thus violates the LPP.

## C.2 VARYING LEARNING TECHNIQUES

Theorem 2 implies that the strict trade-off between a representation's utility for its intended task and the LPP holds regardless of the learning technique used to obtain the feature map $f_E(X) = Z$. In Section 4.2, we show that indeed even with attribute censoring through gradient reversal, an adversary can always find a data attribute for which $\Delta_{PUT} > 0$ and that thus violates the LPP. In this section, we experimentally demonstrate that the same applies to other learning techniques that aim to hide sensitive information about the original data $X$, such as differentially private training or adversarial representation learning.

### C.2.1 DIFFERENTIALLY PRIVATE TRAINING

We train a CNN256 model on the LFWA+ dataset (see Section 4) under differential privacy (Dwork et al., 2014). We use the Opacus library (Yousefpour et al., 2021) to implement simple differentially private stochastic gradient descent with gradient clipping. We test for optimal $\alpha$ in a range of $[2, 32]$ and obtain a final privacy parameter of $\varepsilon = 1.1$.

In Fig. 7, we show that differentially private training limits the worst-case inference gain for any attribute $S$ for records in the training data, but does not prevent test-time attribute inference that LPP aims to prevent. Differentially private training leads to a drop in model performance on the intended task (Fig. 7 (*bottom, right*)) but does not prevent a worst-case adversary that gains an advantage from observing a target record's feature representation.

### C.2.2 ADVERSARIAL REPRESENTATION LEARNING

Zhao et al. (2020) empirically compare the trade-off between hiding sensitive information and task accuracy of various attribute obfuscation algorithms. They find that together with gradient reversal, Maximum Entropy Adversarial Representation Learning (MAX-ENT) provides the best trade-off. We run a simple experiment on the Adult dataset (Kohavi & Becker, 2013) that shows that the trade-off predicted by Theorem 2 also applies to the representations learned by a model trained under MAX-ENT.

**Experiment Setup.** We use the exact same model architecture and data as Zhao et al. (2020). We train the model to predict attribute 'income' and adversaries for four sensitive attributes ('age', 'education', 'race', and 'sex'). We then calculate the utility and inference gain as described in Section 4.

Fig. 8 shows that, as expected, even for a model trained under attribute obfuscation with MAX-ENT, the adversary's inference gain exceeds the model's utility gain for two out of the four sensitive attributes tested. This further supports our theoretical finding that the trade-off between LPP and utility for a prediction task of a representation applies *regardless of how these representations are learned*.

## C.3 VARYING DATASET

We ran an additional experiment to demonstrate that the strict trade-off between model utility and the LPP also holds on a very different type of dataset and model. As for tabular data, together with image data, sharing feature encodings instead of raw data is often suggested as a solution to limit harmful inferences, we choose the Texas Hospital dataset (Texas Department of State Health Services, Austin, Texas, 2013) and the TabNet model architecture (Arik & Pfister, 2021) for these experiments.

**Data.** The Texas Hospital Discharge dataset (Texas Department of State Health Services, Austin, Texas, 2013) is a large public use data file provided by the Texas Department of State Health Services. The dataset we use consists of 5,202,376 records uniformly sampled from a pre-processed data

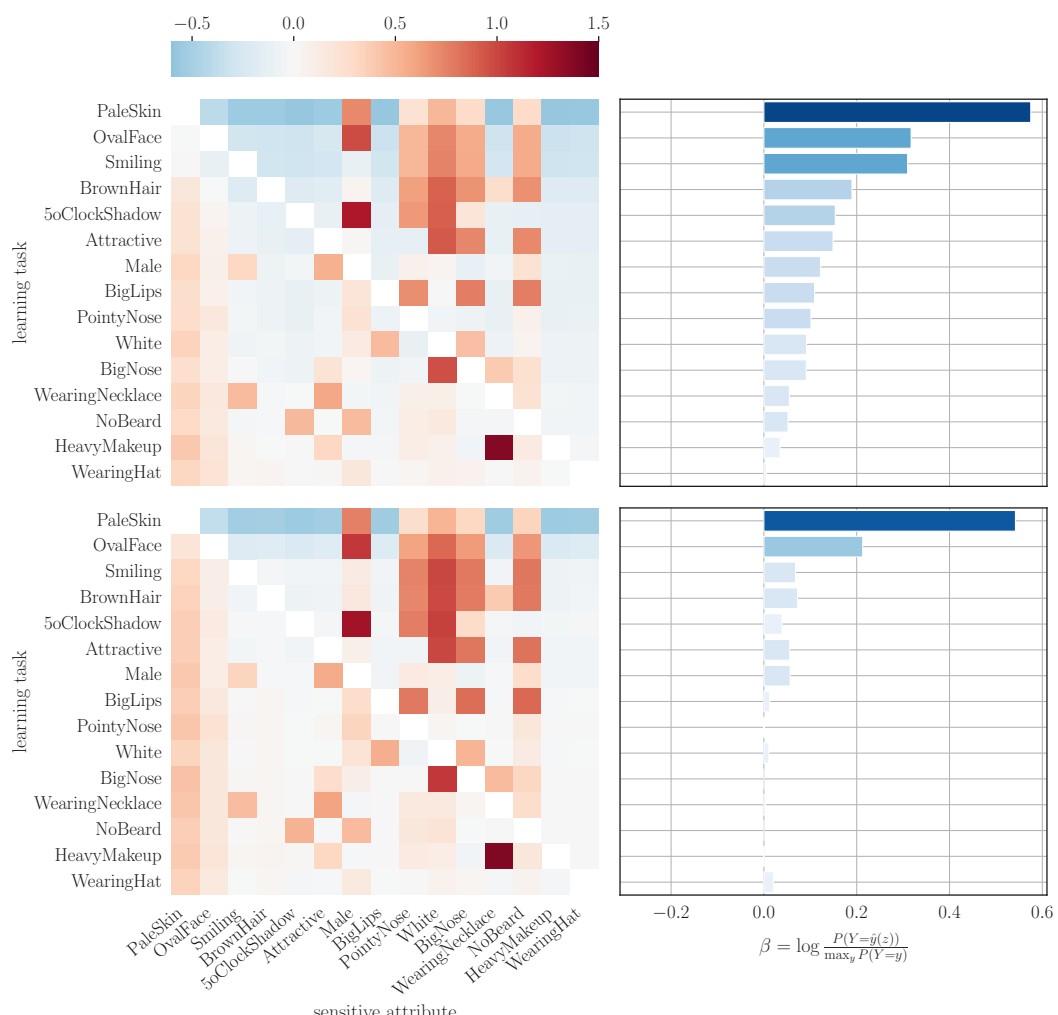

Figure 7: **The least-privilege and utility trade-off holds even under differentially private model training.** The adversary's inference gain (*left*) always exceeds the utility gain (*right*) for at least on sensitive attribute. This hold for both a CNN256 model trained under standard learning techniques (*top*) and under differential privacy (*bottom*).

file that contains patient records from the year 2013. We retain 18 data attributes of which 11 are categorical and 7 continuous.

**Experiment Setup.** In each experiment, we select one attribute as the model's learning task $Y$ and a second attribute as the sensitive attribute $S$ targeted by the adversary. We repeat each experiment 5 times to capture randomness of our measurements for both the model and adversary, and show average results across all 5 repetitions. At the start of the experiment, we split the data into the three sets $D_T$, $D_E$, and $D_A$. We train a TabNet model on the train set $D_T$ for the chosen learning task and then estimate the model's utility on the evaluation set $D_E$. We measure the model's utility by estimating the multiplicative gain $\tilde{I}_\infty(Y; Z) = \log \tilde{\Pr}(Y=\hat{Y}(Z))/\tilde{\Pr}(Y=\hat{Y})$, where $\hat{Y}(Z)$ denotes the trained model's prediction for a record's task label $Y$ and $\hat{Y}$ without the argument the majority class baseline guess. After model training and evaluation, we train both the label-only and features adversary on the auxiliary data $D_A$. The features adversary is given access to a record's representation at the last encoding layer of the TabNet encoder (see Arik & Pfister (2021) for details of the model architecture). For a given sensitive attribute $S$, we estimate the adversary's gain as $\tilde{I}_\infty(S, Z \mid Y) = \log \tilde{\Pr}[S=\hat{S}(Z,Y)]/\tilde{\Pr}[S=\hat{S}(Y)]$.

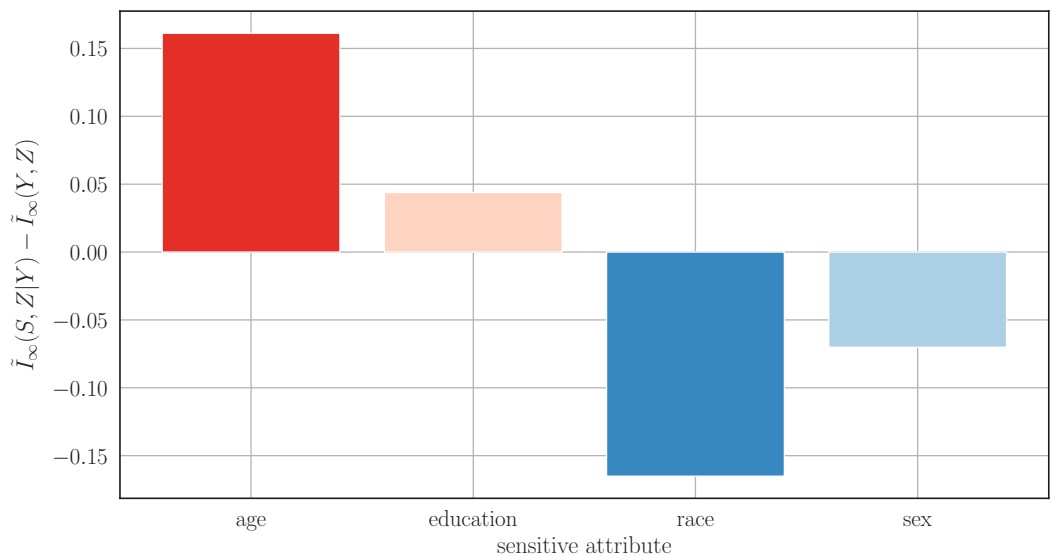

Figure 8: **For two out of four sensitive attributes tested, the adversary's inference gain exceeds the model's utility gain.** We show the delta between the adversary's inference gain and model utility for a model trained under MAX-ENT for learning task 'income' on the Adult dataset.

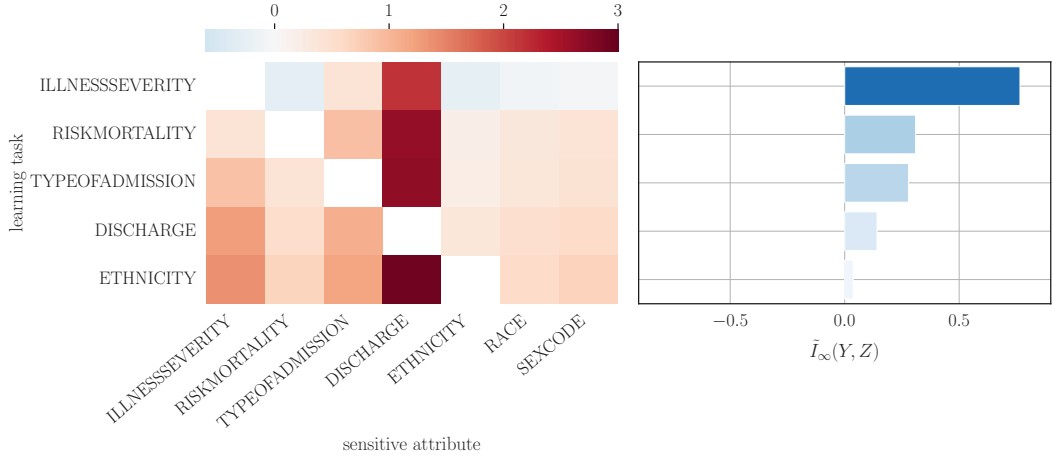

Figure 9: Attribute leakage (*left*) and model utility (*right*) for a TabNet model trained on the Texas Hospital dataset

As above, the bar chart in Fig. 9 (*right*) shows the model's utility for learning task $Y$ indicated in each row measured as $\tilde{I}_\infty(Y, Z)$. The heatmaps in Fig. 9 (*left*) show the difference between the adversary's inference gain and the model's utility $\tilde{I}_\infty(S, Z) - \tilde{I}_\infty(Y, Z)$. As on the LFWA+ dataset, for any learning task there always exists a sensitive attribute for which an adversary gains an advantage from observing a target record's feature representation and $\Delta_{PUT} > 0$. This demonstrates the strict trade-off between utility and the LPP as predicted by Theorem 2.

