# OpenReview forum: "The Fundamental Limits of Least-Privilege Learning"
_ICLR.cc/2024/Conference — Submitted to ICLR 2024_

### Official Review · Reviewer_qV6w · 2023-10-30

**Soundness:** 3 good
**Presentation:** 4 excellent
**Contribution:** 4 excellent
**Rating:** 6
**Confidence:** 3

**Summary:**

This paper studies the data attribute leakage problem in machine learning models. The core idea of this paper is that any representation that provides utility for prediction always leaks information about properties of the data other than the task. This work gives the definition of the least-privilege principle (LPP) and gives the theory of the trade-off between LPP and utility. The work also demonstrates this trade-off experimentally in an image classification setting.

**Strengths:**

This work theoretically analyzes the relationship between data attribute leakage and utility, which is important for the security of machine learning. This work also reveals the inherent properties of machine learning models.

**Weaknesses:**

1. The study of data attribute leakage is an active research field. Some work attempts to mitigate attribute leakage by designing sophisticated algorithms. This paper does not mention and compare cutting-edge defense schemes in the empirical experiment part.

2. More extensive experiments should be used to evaluate the proposed theory. At present, this work is only conducted on a dataset and a neural network model.

**Questions:**

1.	Advanced defense strategies should be discussed. If advanced defense strategies can solve the problem of data attribute leakage, then the value of this work will be limited. Therefore, could the author provide a more adequate overview of the research area discussed? It’s not just data attribute leaks, it should also include cutting-edge defense methods.
2.	Could the authors conduct empirical experiments on a wider range of datasets and models? For example, conduct experiments on some NLP tasks. More extensive experimental results can more fully verify the proposed theory.
3.	Does the complexity of the feature extractor affect the extent of data attribute leakage?
4.	There are some typographical errors in the paper. For example, formulas (12), (14), and (19) in Appendix A lack punctuation.

---

> ### Author Response · Authors · 2023-11-15
>
> Thank you very much for taking the time to read our paper and the valuable suggestions. Below, we try to address your comments.
>
> **Comparison to other learning techniques:** The review rightly notes that the literature suggests a wide range of learning techniques to address attribute leakage in deep network models. As we show in _Section 3.2, Theorem 2,_ however, the trade-off between a representation’s utility and achieving the LPP applies “regardless of the way the feature representations are obtained” (_Section 3.2, page 6_). In Section 4, we provide an empirical evaluation that supports our theoretical results: Regardless of a model's learning task and even under censoring, the LPP cannot be fulfilled while simultaneously providing high utility for the prediction task.
>
> We furthermore note that to the best of our knowledge none of the existing techniques in the literature can actually achieve the desired goal of LPL. To do so, the features learned by the model would need to simultaneously restrict leakage for any sensitive attribute $S$. Techniques to defend against attribute inference are mostly limited to one or two fixed sensitive attributes and hence cannot achieve this goal.
>
> **Complexity of feature extractor:** Our theoretical results imply that the trade-off applies not only regardless of the learning technique, but also regardless of a model’s architecture. We have run additional experiments using a different model architecture (ResNet-18) on the same image dataset to confirm that, as predicted by Theorem 2, the trade-off applies regardless of the exact architecture choices. We have uploaded these results as supplementary material, and we them to the Appendix.
>
> **Additional dataset:** In the supplementary material, we provide the results of an additional experiment which demonstrates the strict trade-off between model utility and the LPP on a very different type of data and model. As for tabular data, together with image data, sharing feature encodings instead of raw data is often suggested as a solution to limit harmful inferences, we chose the Texas Hospital dataset and the TabNet model architecture (Arik & Pfister, 2021) for these experiments.
>
> The results show that, as predicted by Theorem 2 (Section 3.2), for any learning task there always exists a sensitive attribute that violates the LPP.

---

> > ### Author Response · Authors · 2023-11-20
> > **Revised version**
> >
> > We have uploaded a revised version of our paper that contains additional experiment results that confirm that, as predicted by our main theoretical result, the least-privilege and utility trade-off holds for any feature representation. In particular:
> > - In Appendix C.2, we show that the trade-off holds regardless of the learning technique used to obtain the feature map $f_E(X) = Z$. We show results for two more techniques that aim to hide sensitive information about their input data: differentially private training and maximum entropy adversarial representations learning (MAX-ENT). We chose to evaluate the trade-off under MAX-ENT as Zhao et al., 2020 report hat together with gradient reversal (our first choice for previous experiments), MAX-ENT provides the best trade-off between hiding a particular sensitive attribute and task accuracy.
> >  - In Appendix C.1, we show that the trade-off holds regardless of a model's architecture.
> > - In Appendix C.3, we show that the trade-off equally applies to a very different type of data, i.e., on a tabular dataset.
> >
> > We hope that these revisions address your comments in a satisfactory way.
> >
> > [Zhao et al., 2020: https://proceedings.neurips.cc/paper/2020/file/6b8b8e3bd6ad94b985c1b1f1b7a94cb2-Paper.pdf]

---

### Official Review · Reviewer_KceC · 2023-11-01

**Soundness:** 3 good
**Presentation:** 3 good
**Contribution:** 3 good
**Rating:** 5
**Confidence:** 4

**Summary:**

The paper studies unintended privacy leakage in collaborative learning. Specifically, the paper proposes to  formalize of the least-privilege principle for machine learning. Via information theory, the paper observes that every task comes with fundamental leakage—a representation shared for a particular task must reveal the information that can be inferred from the task label itself. Such fundamental leakage is also testified in a real-world dataset

**Strengths:**

+The paper is well-written and easy to follow

+Understanding the fundamental information leakage in collaborative learning is important

**Weaknesses:**

-The key differences with the existing information-theoretic privacy is unclear

-The observations are only shown on a single dataset

**Questions:**

While the paper uses least-privilege principle to formalize information leakage, I do not know how this largely differentiate other works that use the similar idea (though those works assume a fixed sensitive attribute), e.g., Zhao et al., 2020; Brown et al., 2022 and Salamatian et al., Privacy-Utility Tradeoff and Privacy Funnel. What is the key technical challenge when we do not assume a fixed sensitive attribute, but assume it is a superset of the input?

Theoretically, the paper shows the fundamental leakage and observes this on a dataset. I am curious how common such observation is in more datasets.

The evaluation is only tested on a single attribute inference. How generalizable it is to more attribute inference (e.g., data reconstruction attack)?

---

> ### Author Response · Authors · 2023-11-15
>
> Thank you very much for the review and your questions which we try to clarify below.
>
> **Key difference to existing information-theoretic approaches:**
>
> _Connection to CEB:_ Our formalisation of the (previously only stated informally) problem of least-privilege learning can be thought of as a version of the generalised Conditional Entropy Bottleneck (CEB) problem (Fischer, 2020), as we explain in Section 3.2, page 5. In contrast to standard CEB, we consider a stronger notion of leakage, i.e., maximal leakage rather than mutual information, between the representations $Z$ and the sensitive raw data $X$. Weaker notions of leakage are not adequate in our case where we want to assess the least-privilege claim.
>
> _Connection to Privacy Funnels:_ The Privacy Funnel (PF) problem is different from the least-privilege problem that we consider in this work; as Asoodeh & Calmon, 2020 point out, PF and bottleneck-type problems like CEB are "duals" of each other.
>
> As the review notes, privacy funnels operate in a different setting. Most crucially, because they consider only a single fixed attribute. As we discuss in Section 3 (“Attribute Inference”), we have to treat any attribute other than the learning task as sensitive to capture the _least-privilege_ principle: We want the representation to leak _nothing_ else other than the fundamental leakage implied by the learning task.
>
> **Limits of empirical evaluation:** In the Supplementary Material, we provide the results of an additional experiment which demonstrates that the strict trade-off between model utility and the LPP also holds on a very different type of dataset and model. As for tabular data, together with image data, sharing feature encodings instead of raw data is often suggested as a solution to limit harmful inferences, we chose the Texas Hospital dataset and the TabNet model architecture (Arik & Pfister, 2021) for these experiments.
>
> The results show that, as predicted by Theorem 2, Section 3.2, for any learning task there always exists a sensitive attribute that violates the LPL.
>
> We limit the empirical evaluation to single attribute inference attacks as, despite their simplicity, they are sufficient to support our theoretical result: As soon as the adversary’s inference gain on a single attribute exceeds the features’ utility for the learning task, the features cannot be claimed to fulfil the LPL claim. We note, however, that Theorem 2 covers any type of inference attack, incl. data reconstruction attacks for the case where $S=X$.

---

> > ### Comment · Reviewer_KceC · 2023-11-21
> > **Reply to Authors' Response**
> >
> > Thanks for the comments!
> >
> > However, I am still unclear to the novelty of the paper and the rationality of the assumption.
> >
> > First, as pointed out by the authors, the survey paper [a] already well clarifies the terminology, such as information bottleneck and privacy funnel, which are introduced in the form of mutual information, entropy.  In addition, [b] has theoretically analyzed the Fundamental limits of perfect privacy, while the maximal leakage concept and formulation are from [c]. Compared with these prior work, whats the key differences of the proposed theoretical results? What are the challenges? I checked the proof, but did not see new results? Please correct me if I am wrong.
> >
> > Second, the authors also assume that the space of inputs is a subset of the space of sensitive attributes. What real-world scenarios make this assumption feasible? Is this only for the purpose of deriving the theoretical results (e.g., inspired by Theorem 1 in [b])? Note that the evaluation only tests on a single private attribute, which did not satisfy the assumption.
> >
> > [a] Asoodeh and Calmon, Bottleneck Problems: Information and Estimation-Theoretic View
> >
> > [b] Calmon and Me ́dard, Fundamental limits of perfect privacy, 2015.
> >
> > [c] Issa et al. An Operational Approach to Information Leakage. IEEE TRANSACTIONS ON INFORMATION THEORY,  2020

---

> > > ### Author Response · Authors · 2023-11-22
> > >
> > > **Novelty**: Our proofs leverage known mathematical arguments and prior information theoretic results; and we believe that this is clearly expressed in the manuscript via appropriate references.
> > > However, our main result (Theorem 2, Section 3.2) **is a new formal result about a novel problem** that has not been stated in any prior work nor can be derived as a corollary from any of the related information-theoretic problems.
> > >
> > > Reference [b] characterises the fundamental limits of the Privacy Funnel (PF) problem. In PF, motivated by the problem of privacy-preserving data publishing or anonymisation, the goal is to publish a representation $Z$ of the original data $X$ that preserves at least $t$ bits of information about $X$, i.e., $I(Z,X) > t$ while minimising the information about a sensitive attribute $S$ with $I(Z, S)$. A crucial difference between the PF problem and the least-privilege problem that we study is that in PF there exists no definition of the *intended learning task* critical to the least-privilege concept  (represented in our work through the task label $Y$ and reflected in the utility definition $I(Z ,Y)$). Other differences are that PF only considers _a fixed set of sensitive attributes_ instead of all attributes other than the task and uses a weaker notion of leakage (mutual information compared to maximal leakage).
> > >
> > > As we discuss in Section 3.2 (page 5 and 6), the least-privilege problem that we study is most closely related to information bottleneck problems, such as CEB. However, there are crucial differences to any of the previously studied problems. For instance, to properly formalise the least-privilege principle – to learn nothing else other than the intended task – we need to consider a stronger notion of leakage than mutual information as in CEB.
> > >
> > > In our work, we show that these differences between the least-privilege and other problems (PF or CEB) lead to a very different trade-off, **which was not known prior to our work**.
> > >
> > > **Sensitive attributes**: We consider the space of inputs as a subset of the space of sensitive attributes to capture the risk of (partial) data reconstruction attacks; which would, like single attribute inference attacks, lead to harmful inferences and violate the least-privilege principle. In our empirical evaluation, we only test single attribute inference attacks as these are sufficient to support our theoretical results: We demonstrate that for any feature map, learning task, and training procedure there always exists at least one sensitive attribute that violates the least-privilege principle. As we show in Appendix C.3, this also holds for tabular data where the chosen sensitive attribute is a part of the input record.

---

> > > ### Author Response · Authors · 2023-11-22
> > >
> > > Thank you for engaging with us during this discussion period! Please let us know if there is anything else that we could clarify. Given that the revised version includes additional experimental results on a tabular dataset as well as results with other model architectures and training methods, and in the case we have clarified the concerns regarding the differences to the related literature, we were wondering if you would consider raising the score.

---

> ### Author Response · Authors · 2023-11-20
> **Revised version**
>
> We have uploaded a revised version of our paper that
> - clarifies the connection to related information theoretic problems, such as CEB and Privacy Funnel (see Section 3.2 page 5).
> - contains additional experiment results that confirm that, as predicted by our theoretical results, the trade-off also applies to other types of data (see Appendix C.3, Figure 9).
>
> We hope that these revisions address your comments in a satisfactory way.

---

### Official Review · Reviewer_kXjd · 2023-11-02

**Soundness:** 3 good
**Presentation:** 2 fair
**Contribution:** 2 fair
**Rating:** 5
**Confidence:** 4

**Summary:**

This contribution addresses the concerns of data misuse when offloading model training and inference to a service provider. Collaborative learning and model partitioning are proposed as solutions, where clients share representations of their data instead of the raw data. The principle of least privilege is introduced, which states that the shared representations should only include information relevant to the task at hand. The authors provide the first formalization of the least-privilege principle for machine learning. They prove that there is a trade-off between the utility of the representations and the leakage of information beyond the task. Experiments on image classification demonstrate that representations with good utility also leak more information about the original data than the task label itself. As a result, censoring techniques that hide specific data attributes cannot achieve the goal of least-privilege learning.

**Strengths:**

This paper reveals the fundamental limits of Least-Privilege learning.

**Weaknesses:**

1. The presentaion needs improvement.

2. The contribution is limited.

**Questions:**

See above.

---

> ### Author Response · Authors · 2023-11-15
>
> Thank you for the review. As the review has pointed out in the summary, our paper provides the first formalisation and a generic characterization of the trade-offs involved in least-privilege learning, which was proposed in the prior work (Melis et al., 2019; Brown et al., 2022). Our characterization applies to any method of obtaining feature representations, covering the model partitioning and collaborative learning settings. Please let us know if there are any questions we can answer to change the assessment of the contribution significance, or to clarify any concrete issues with the presentation.

---

### Official Review · Reviewer_3Rfy · 2023-11-06

**Soundness:** 3 good
**Presentation:** 3 good
**Contribution:** 2 fair
**Rating:** 6
**Confidence:** 3

**Summary:**

This work investigates the limits to information leakage by considering the least privilege principle. It formalises this notion under strict condition where any attribute other than the task label is deemed as sensitive for leakage. They propose a formal definition of LPP and under strictly positive posterior and the assumption that the label needs to be shared with the service provider, they theorise that there does not exists a a feature map such that both LPP and utility (as defined by mutual information between labels and feature representation) hold simultaneously.  They further support this with pairwise empirical evaluations across 12 attributes where one is considered as label and the other as the adversary’s targeted sensitive attribute. The analysis also compares inference gain under standard and censoring models.

In my opinion the paper relies on some fundamental assumptions which have been clearly stated
- First, it’s a worst case analysis. This has also been highlighted in the Problem Setup (Section 2) and the text following Corollary 1.
- I think the assumption for label information to be made available is pretty strong from a practical perspective as users may not necessarily need to provide labels to the service provider.
- Unlike unconditional LPP, the LPP is defined with respect to what the authors consider as fundamental limit of leakage from the label.

**Strengths:**

I found the paper to be generally well written and easy to follow. While I am not familiar with all the current literature in this field, I found that the paper clearly states its worst-case assumptions and conveys its theoretical results with insights.

**Weaknesses:**

I found the discussion on how users can manage the trade-offs to be quite limited. There is some discussion on DP and its usefulness for training but not test time inference but I think some more discussion on what this theoretical analysis would mean for a user would be useful.

**Questions:**

Although I am not fully familiar with the related literature on this, I found the paper to be self-contained with insightful discussion.
I voted for marginal acceptance because of the following, and would appreciate if authors can help clarify

1. Practical implications for users and service providers because of the noted limits
2. Arguments for why sharing label information is a practical worst-case assumption
3. How the work relates to privacy-preserving techniques other than censoring, like de-anonymisation or sharing the information under encryption etc? While I understand this may not be within the scope and/or page limit constraints, I think even qualitative arguments can help position the paper for discussion within the privacy and ML community.

---

> ### Author Response · Authors · 2023-11-15
>
> We would like to thank you for the thoughtful review and the questions which we will try to address as best as possible next.
>
> **Practical implications for users and service providers:** This is an excellent question which we tried to partially address in Sections 4.1 and 4.2 (“Takeaways”). The primary conclusion we draw from our results is that service providers should carefully evaluate the trade-off between utility and unintended information leakage for a given learning task and more openly communicate their results to increase transparency and avoid unexpected inferences, with potentially harmful consequences, on the user side. In particular, service providers should evaluate the fundamental leakage of a task and inform users that “any data attribute that is correlated with the chosen task” (Section 4.1, page 8) might also be revealed. This would enable users to make informed choices about the risk they might take by the use of a prediction service or by contributing their data for model training.
>
> Second, going forward, we encourage both researchers and practitioners to more rigorously evaluate broad claims, such as, the claim that sharing feature representations instead of raw data records limits information leakage to “only the features relevant to a given task” (Melis et al., 2019). We hope that our formal definitions serve as a good basis to evaluate such claims and any future proposals.
>
> **Strong assumption on sharing label information:** The assumption that the service provider has access to the correct label information of a particular target record is primarily motivated by the observation that it is impossible to achieve the strict LPP definition given in Section 3.1, Definition 1. To allow for any meaningful learning, we have to at least allow for the fundamental leakage through the task label (see _Section 3.2, page 5_), which means we have to assume that the label is known to the adversary. We note that a weaker assumption on the fundamental leakage adversary would result in an even worse trade-off between unintended feature leakage and a model’s utility.
>
> In some preliminary experiments, we did assess the success of an adversary that only has access to a record’s predicted task label instead of the ground truth. We observed that, as expected, the adversary’s success in predicting a sensitive attribute from the predicted task label increased with increasing model utility. In our analysis and final experiments, we decided to only consider the worst-case adversary that has access to the correct task label as this gives us a lower bound on the adversary’s multiplicative gain and hence the best method to analyse the trade-off.
>
> **Relation to other privacy-preserving techniques:** ln short, alternative solutions to this problem, such as encryption or complete data isolation and local processing are orthogonal to the idea of sharing feature representations instead of the raw data to prevent harmful inferences. As Osia et al., 2017 and 2018, discuss, these methods are often not efficient enough for big data and deep learning.
>
> Data anonymisation can indeed be thought of as an instance of sharing a representation of the data rather than the data itself (see [Makhdoumi, et al., 2014](https://arxiv.org/abs/1402.1774)); with the crucial difference that the set of sensitive attributes is fixed and limited and that the learning task is not defined upfront. This hence leads to a different trade-off between preserving and hiding information than in our setting.

---

### Official Review · Reviewer_orQ6 · 2023-11-09

**Soundness:** 3 good
**Presentation:** 2 fair
**Contribution:** 2 fair
**Rating:** 5
**Confidence:** 3

**Summary:**

The paper analyzes inference-time information leakage due to releasing the representations of sensitive input data (compared to only releasing the predicted label). It investigates how this leakage relates to the quality of the representation. For modeling the quality of representation (concerning a downstream prediction task), the authors analyze the mutual information between the representation and the label. For modeling the information leakage from the representation about the sensitive attributes, the authors analyze the Bayesian optimal adversary for attribute inference. The authors prove that whenever the representation enables non-negligible performance for downstream prediction tasks, there must exist attributes that are significantly more leaked through the additional release of representation (compared to only releasing the label) of the sensitive input data.

To interpret and validate this inherent trade-off, the authors further perform experiments on tabular datasets to quantify the empirical information leakage and model utility. Specifically, information leakage is measured against an empirically instantiated Bayesian optimal adversary using auxiliary data.

**Strengths:**

- The authors prove an interesting inherent trade-off between the model's utility and the information leakage due to releasing representation (compared to only releasing labels). As an interesting baseline, the authors also discussed the fundamental information leakage of how much the label of input data reveals sensitive attributes.

- Experiments support the proved trade-off, as the authors observe a positive correlation between the number of attributes that incur high information leakage (due to releasing representations) and the model's utility. Interestingly, when the most leaked attributes are censored during the model training and inference phase, the authors observe that the leakage about other attributes increases.

**Weaknesses:**

- The model splitting between server and clients analyzed in this paper is counter-intuitive. Namely, the authors assume that clients use only the representation layers of the model, while the server uses only a classification head of the model. Since the classification heads are usually small and easy to train, I do not see the incentive for clients to share input data representations to the server in this setting (for prediction tasks). For example, the clients may download the classification head weights or tune a classification head on local data and compute the labels locally at inference time. This also seems different from Melis et al. 2019, where the clients only compute one embedding layer (rather than a large part of the whole model).

- The definitions and notations lack clarity at times. Most importantly, the information leakage is defined by successful inference of *any* attribute. This may be overly strong as many attributes are less sensitive. I'm wondering whether the proved trade-off in this paper is largely a result of this overly strong definition of information leakage.

**Questions:**

- Could the authors discuss more about the effect of the splitting method, i.e., what parts are deemed as representations, on the theoretical and empirical conclusions? For example, when the client only has access to a small part of the model, would the leakage still be unbalanced on different attributes?

- Could the authors discuss how the theoretical and empirical trade-off (between information leakage and model utility) might change if we only consider a smaller set of attributes (rather than the whole input feature space) as sensitive?

Minor comments regarding clarity:
- What are the transition orders between random variables in the Markov chain $Y - X - W$ defined in Section 2?
- In Figure 4, bottom right plot, what is the meaning of color? Why is the third row from the top colored blue despite its negative value? What does a negative model utility $\tilde{I}_{\infty}(Y, Z)$ mean?
- Assumption A requires a positive posterior but does not require any lower bound for the posterior density. Does it mean the posterior could also be arbitrarily close to a point distribution?

---

> ### Author Response · Authors · 2023-11-15
> **Responses to the major comments and questions**
>
> First, thank you for taking the time to review our work and the insightful comments which we try to address below.
>
> **Model splitting between server and clients is counter-intuitive:** The model partitioning setup, as we describe it in this paper, is a common paradigm in the area of privacy-preserving cloud computing and in the context of privacy for MLaaS (see, for instance, Osia et al., 2018, Wang et al., 2018, Chi et al., 2018, Li et al., 2021, and Brown et al., 2022). The main motivation, cited in all works and also the focus of our work, is to prevent harmful inferences from raw data records. Other reasons to not share the full model might include intellectual property concerns or scalability considerations (see Osia et al., 2018 for a more detailed discussion).
>
> **Effect of the splitting method on theoretical and empirical conclusions:** In _Section 3.2, Theorem 2,_ we show that the trade-off between a representation’s utility and achieving the LPP applies to any feature representation regardless of the exact splitting method, model architecture or “the way the feature representations are obtained” (_Section 3.2, page 6_).
>
> In our experiments, we evaluate the trade-off between utility and unintended inferences on the last layer feature representations of a CNN model. We choose the last layer because it is where a record’s feature activations are expected to be most learning task specific (Melis et al., 2019 and Mo et al. 2021) and hence the claim that these representations meet the LPP is the strongest. Our results demonstrate that, even at the last layer and under censoring of particular attributes, the least-privilege claim does not hold true.
>
> We have run additional experiments: (1) using a different model architecture (ResNet-18) on the same image dataset, and (2) using a different model (TabNet) on an additional tabular dataset (Texas Hospital dataset) to confirm that, as predicted by Theorem 2, the trade-off applies regardless of the exact model architecture choices. We have uploaded these empirical results as supplementary material, and will add add them to the Appendix.
>
> **Difference to Melis et al.:** The review points out correctly that the setting which we use to motivate our work and in our empirical evaluation differs from the setting described by Melis et al., 2019 who model unintended feature leakage in a collaborative learning setting. Yet, as we state at the beginning of Section 2: “all of our formal results apply to any setting in which feature representations are used as a means to limit the data revealed to untrusted third parties, such as the collaborative learning setting” (_Section 2, page 2_). The gradients shared during collaborative learning are a noisy representation of the features learned by the model, and hence are also captured by our setting $Z = f_E(X)$ in the same way as feature representations in the model partitioning scenario.
>
>
> **Worst-case assumption on attributes considered sensitive and how it affects the trade-off:** As mentioned in Section 3 ("Attribute Inference"), we want to model an adversary that captures the _least-privilege_ principle which implies that the representations shared with the service provider leak _nothing else_ other than the fundamental leakage implied by the learning task. To adequately capture this principle, we have to consider any attribute, other than the learning task, to be sensitive.
>
> If we were to restrict the set of attributes considered sensitive, the problem would become similar to a version of the Privacy Funnel (PF) problem (Asoodeh & Calmon’20). PF is different from the least-privilege learning problem (a form of generalized Conditional Entropy Bottleneck (CEB) problem; see next) that we consider in this work. As Asoodeh & Calmon point out, PF and bottleneck-type problems like CEB  are "duals" of each other.
>
> **Trade-off as a result of strong assumptions:** As we discuss in Section 3.2, the strict trade-off between utility and the LPP exists because the LPP restricts the adversary’s inference gain in terms of maximal leakage. In the standard CEB problem (Fischer, 2020), which considers a weaker notion of unintended leakage, the mutual information between the representations $Z$ and the sensitive raw data $X$, there is hence no such trade-off (see _Section 3.2, page 6_). Such a weak notion of leakage, however, is not adequate to formalise and assess the original claim from many papers that it is possible to learn feature representations that protect against any harmful inferences and fulfil the least-privilege principle.

---

> ### Author Response · Authors · 2023-11-16
> **Response to minor comments**
>
> **Response to minor comments:**
>
> - The transition kernels between Y - X - W can be arbitrary so long as $Y \bot W \mid X$.
>
> - The positive posterior requirement implies both upper and lower bound as the notation is a shorthand for $P(Y = y \mid X = x) > 0$ for all $y \in \mathbb{Y}$. Indeed, the assumption allows distributions that are arbitrarily close to a point distribution, but not exactly point distributions.
>
> - The bar in the third row should indeed be coloured red. We apologise for this data plotting error. A negative value indicates that the performance of the classifier has dropped below the majority class baseline guess, i.e., that the features learned by the model do not provide any utility for the prediction task. This is a common side effect of censoring techniques which often lead to a degradation in model performance (see, for instance, Song & Shmatikov, 2019 and Raff & Sylvester, 2018).

---

> ### Author Response · Authors · 2023-11-20
> **New revision**
>
> We have uploaded a revised version of our paper that
> - better justifies our theoretical assumptions to consider all attributes as sensitive and to analyse the trade-off in terms of maximal leakage (see Section 3.2 page 5).
> - better explains how these theoretical assumptions affect the strict trade-off between the least-privilege principle and utility (see Section 3.2 page 6)
> - contains additional experiment results that confirm that, as predicted by our theoretical results, the trade-off applies to any feature representation regardless of the feature mapping (see Appendix C.1 and C.3, Figures 6 and 9).
> - contains an updated version of Figure 4 that fixes the data plotting error that this reviewer thankfully pointed out.
>
> We hope that these revisions address your comments in a satisfactory way.

---

> ### Comment · Reviewer_orQ6 · 2023-11-20
>
> Thanks to the authors for the response and clarifications. The evaluated partitioning scheme (where the server only has a small part of the model) and the overly strong privacy definition still lack significance to me.
>
> Re the model partitioning scheme, let me rephrase the discrepancy. The authors only evaluate settings with a **small** classifier on the server side. By contrast, the model on the server side is **large** in most (if not all) cited works, especially in Melis et al.
>
> Re the significance of the trade-off, as the authors acknowledge, the results largely result from the strong privacy definition and do not easily extend to settings with a subset of sensitive attributes. This makes the results less interesting/useful, as they only reinforce a common belief in the early work of Melis et al. below.
> > Of course, the purpose of ML is to discover new information about the data. Any useful ML model reveals something about the population from which the training data was drawn. For example, in addition to accurately classifying its inputs, a classifier model may reveal the features that characterize a given class or help construct data points that belong to this class. In this paper, we focus on inferring "unintended" features, i.e., properties that hold for certain subsets of the training data, but not generically for all class members.
>
> This belief is the **whole motivation** of Melis et al., and follow-up works to focus on unintended features rather than all features. Therefore, why should we care about all features again in this paper?

---

> ### Comment · Reviewer_orQ6 · 2023-11-20
> **Additional question regarding Theorem 2**
>
> Additionally, I'm puzzled by whether/how the trade-off under LPP (Theorem 2) does not hold under the conceptual representation $Z=\arg\max_{y}P(Y=y|X)$. Specifically, this representation does not reveal any more information than the class of the input data $X$, yet it yields Bayes optimal classification success.
> - Is it because it violates certain assumptions required for Theorem 2?
> - If so, does it imply Theorem 2 is too limited to cover representations that are similarly good, i.e., achieve Bayes-optimal classification success?

---

> ### Author Response · Authors · 2023-11-22
>
> **Regarding the model partitioning scheme:** To clarify this point, we would like to distinguish between two questions  (1) In what form are data representations made available to the service provider/adversary? (2) What is the architecture of the model and how is it split into an encoding $f_E$ and classification part $f_C$?
>
> On (1), as we describe at the start of Section 2, some works evaluate unintended feature leakage in a collaborative learning setting (Melis et al., 2019), others in  a model partitioning setting (Brown et al., 2020, Song & Shmatikov, 2020, Zhao et al., 2020 or Osia et al., 2018).
>
> Melis et al., 2019 assume that the adversary/service provider has access to _the full set of gradient updates across all layers of the model_. They state that these "gradient updates can [...] be used to infer feature values, which are in turn based on [...] private training data". In our notation, the data representations $Z = f_E(X)$ are hence the (noisy) feature values of a (batch of) training examples across _all layers of the model_.
> Works such as Brown et al., 2020, Song & Shmatikov, 2020 or Zhao et al., 2020 assume that the service provider is given direct access to a record's representation at a _particular output layer of the model_.
>
> While we have empirically evaluated the model partitioning setting, we highlight that **our theoretical results equally apply to both settings.**
>
> On (2), this question only becomes relevant in the model partitioning setting. Here, we decide at which layer we split a model into its encoding and classification parts. This decision determines the output of the encoder $Z = f_E(X)$, which in turn determines what information is made available to the service provider/adversary to make a prediction about $Y = f_C(Z)$ and to makes a guess about $S = g(Z)$.
>
> The main result of our work (Theorem 2) shows that the trade-off between limiting the inference about $S$ and preserving information about $Y$ holds for _any feature mapping $f_E$_; and equally for any classifier $f_C(Z)$. That is, **these results hold independently of the size of the classification part on the server side**.
>
> To provide evidence of this fact, in Appendix C.2.2, we added an experiment in which we used the code by Zhao et al., 2020 to reproduce our results under their model architecture and training method (maximum entropy adversarial representation learning). In Figure 8, we show that the trade-off in Theorem 2 also applies to this scenario, which has a more extensive classification part than our original experiments.
>
>
> **Regarding the assumptions and significance of the results**: The goal of this paper is to "provide [a] formalisation of the least-privilege principle for machine learning" (Section 1, page 2). The least-privilege concept is repeatedly mentioned in previous works, including Melis et al., as a promising avenue to solve the unintended leakage problem:
>
> "We [...] showed that [...] defenses [...] are not effective. This should motivate future work on better defenses. Techniques that learn only the features relevant to a given task can potentially serve as the basis of "least-privilege" collaboratively trained models." (Melis et al., 2019, Section XI "Conclusions").
>
> Even though many works point to  the least-privilege concept as a solution, "so far this idea has not been formalised. Our goal is to formalise this principle and characterise its feasibility." (Section 2, page 3).
> Our formalisation allows us to show, for the first time, that the least-privilege principle turns out to be elusive in representation learning. Our results demonstrate that, as the reviewer suspects, the assumption that it is possible to achieve least-privilege learning, "learn only the features relevant to a given task" (Melis et al., 2019),  is “too strong”. However, **this was not at all obvious prior to our result.** Instead, prior work repeatedly called for least-privilege learning as a promising way to solve the problem of overlearning (see, for instance, Melis et al., 2019, Brown et al., 2020).
>
> As least-privilege learning comes with a stringent trade off, we agree with the reviewer’s sentiment that the only feasible goal is to limit inference with respect to a specific set of attributes. At that point, however, we have to abandon the concept of least-privilege learning as in “learning representations that contain information that is only relevant to the task itself” and move towards attribute obfuscation and formalisations such as the privacy funnel (see Zhao et al., 2020, Asoodeh and Calmon, 2020 for trade-offs for these problems). How to do that and what are its fundamental limits, is simply a different problem than the one we aimed to formalise and characterise.

---

> ### Author Response · Authors · 2023-11-22
> **Response to question about Theorem 2**
>
> We agree that this instance of $Z$ can seem counter-intuitive, but it follows the trade-off just like any other feature representation. To see this, observe that this instance of $Z$ can actually reveal more information about $X$ than only the class label $Y$. It is easiest to see with an example. Consider a binary input space $\mathbb{X} = \{a, b\}$, and binary label space $\mathbb{Y} = \{0, 1\}$, both uniformly distributed: $P(Y = 1) = \frac{1}{2}$ and $P(X = a) = \frac{1}{2}$. Assume the data distribution is as follows:
>
> |   $X$   | $P(Y = 0 \mid X)$ | $P(Y = 1 \mid X)$ | $P(Z = 0 \mid X)$ | $P(Z = 1 \mid X)$ |
> |:-------:|:----------:|:----------:|:----------:|:----------:|
> |   $a$   |     0.6    |     0.4    |     1      |     0      |
> |   $b$   |     0.4    |     0.6    |     0      |     1      |
>
>
>
> Consider a full data reconstruction attack, so the sensitive attribute is $S = X$. Observe that optimal accuracy of predicting $X$ from only class label $Y$ is $P(X = \hat X(Y)) = \frac{1}{2} \cdot 0.6 + \frac{1}{2} \cdot 0.6 = 0.6$, so 60%. At the same time, observe that $Z$ and $X$ in this case are one-to-one, as $Z$ is a deterministic function of $X$. So, accuracy of predicting $X$ from $(Z, Y)$ is $P(X = \hat X(Z, Y)) \geq P(X = \hat X(Z)) = 1$, so 100%! Therefore, multiplicative leakage is quite high, $\log \frac{1}{0.6} \approx 0.74$. This is because $Y$ is random, so predicting $X$ from $Y$ carries uncertainty, whereas this instance of $Z$ always reveals $X$ exactly.

---

> > ### Comment · Reviewer_orQ6 · 2023-11-22
> >
> > Thanks for the response. I am not quite convinced that the trade-offs still hold under the conceptual representation example. The authors gave specific construction where releasing this representation Z leaks more information about input X than the leakage for only releasing label Y. However, this construction seems more like an artifact because the mapping between X and Z is a bijection. What if multiple values of X map to the same representation value Z (e.g., many inputs X have the same label distribution)? What if the label distributions are closer to point distributions (under which this conceptual representation's performance seems to improve and its induced information leakage decreases simultaneously)? I do not see how the current trade-off Theorem 2 could hold without change under all these cases.
> >
> > Even if Theorem 2 holds under this conceptual representation, this quite counterintuitive example raises additional doubts about how the leakage metric in Theorem 2 is meaningful and how serious the consequence of the proved trade-off is.

---

> > > ### Author Response · Authors · 2023-11-23
> > >
> > > That is indeed a very counter-intuitive case, so we totally get the confusion! Let us generalize the table from the previous example to allow for distributions of $P(Y | X)$ that are arbitrarily close to point distributions as your comment suggests:
> > >
> > > |   $X$   | $P(Y = 0 \mid X)$ | $P(Y = 1 \mid X)$ | $P(Z = 0 \mid X)$ | $P(Z = 1 \mid X)$ |
> > > |:-------:|:----------:|:----------:|:----------:|:----------:|
> > > |   $a$   |  $1 - \varepsilon$	| $\varepsilon$   	| 	1  	| 	0  	|
> > > |   $b$   |  $\varepsilon$     	| $1 - \varepsilon$ | 	0  	| 	1  	|
> > >
> > > where $0 < \varepsilon < \frac{1}{2}$. Note that compared to the previous setup, we change $0.6$ to $1 - \varepsilon$ and $0.4$ to $\varepsilon$. By setting $\varepsilon \approx 0$, we get arbitrarily close to the point distribution.
> > >
> > > To see that the trade-off holds in this case, let us fully work out the utility term $I(Y, Z)$. To compute the mutual information, observe that $P(Y, Z)$ is as follows:
> > >
> > > | $Y$ | $Z$ | $P(Y, Z)$ |
> > > |:---:|:---:|:------:|
> > > | 0 | 0 | $\frac{1 - \varepsilon}{2}$ |
> > > | 0 | 1 | $\frac{\varepsilon}{2}$ |
> > > | 1 | 0 | $\frac{\varepsilon}{2}$ |
> > > | 1 | 1 | $\frac{1 - \varepsilon}{2}$ |
> > >
> > > So, $H(Y | Z) = - (1 - \varepsilon) \log(1 - \varepsilon) - \varepsilon \log(\varepsilon)$, and $I(Y, Z) = \log |\mathbb{Y}| - H(Y | Z) = 1 + (1 - \varepsilon) \log(1 - \varepsilon) + \varepsilon \log(\varepsilon)$. Observe that utility $I(Y, Z) = 1$ when $\varepsilon = 0$, and $I(Y, Z) = 0$ when $\varepsilon = \frac{1}{2}$ (see [plot](https://www.wolframalpha.com/input?i=plot+1+%2B+%281-epsilon%29+log2%281-epsilon%29+%2B+epsilon+log2%28epsilon%29+for+0+%3C+epsilon+%3C+1%2F2)). Importantly, for $0 < \varepsilon < \frac{1}{2}$ the utility in this example is bounded: $I(Y; Z) \leq 1$.
> > >
> > > Next, to get the second part of the trade-off, let us also compute attribute inference advantage. For this, we can directly compute it via the closed form of maximal leakage after applying the Markov chain of our setup (see Eq. 23 in the Appendix). We have $\mathcal{L}(X \rightarrow Z \mid Y) = \log(\max_{y \in \{0, 1\}} \sum_{z \in \{0, 1\}} \max_{x \in \{a, b\}} P(z \mid x)) = \log(1 + 1) = 1$.
> > > Thus, the utility term is smaller than the inference advantage, $I(Y; Z) \leq \mathcal{L}(X \rightarrow Z \mid Y) = 1$ for any $0 < \varepsilon < \frac{1}{2}$, which is consistent with our result.
> > >
> > > Why is this the case? For $P(Y | X)$ that are close to point distributions, the Bayes-optimal instantiation of $Z$ is distributionally very similar to $Y$.
> > > One might thus assume that the utility of predicting $Y$ from $Z$ is high, whereas leakage is low.
> > > As we showed, however, this $Z$ is actually very revealing about $X$ but does not give us perfect information for the prediction task!
> > > In the exact calculations above we see that, in fact, $Z$ has maximum leakage $\log |\mathbb{X}| = 1$ and does not achieve perfect prediction accuracy which corresponds to the utility term $I(Y, Z)$ always being [slightly less than 1](https://www.wolframalpha.com/input?i=plot+1+%2B+%281-epsilon%29+log2%281-epsilon%29+%2B+epsilon+log2%28epsilon%29+for+0+%3C+epsilon+%3C+1%2F2) for any $\varepsilon > 0$.
> > >
> > > We hope this clarifies the confusion with this specific instantiation of $Z$.

---

> > > > ### Comment · Reviewer_orQ6 · 2023-12-04
> > > >
> > > > Thanks for the response. I've read the explanations in details, but remain doubtful about the correctness of the proven trade-off. By definition (3), the leakage gain of observing representation $Z$ (compared to observing label $Y$) is $I_{\infty}(X, Z|Y) = \log(\frac{Pr(X = \hat{X}(Z, Y))}{Pr(X = \hat{X}(Y))})$, where $\hat{S}$ denotes the Bayes optimal adversary. Under point label distribution (i.e., $\varepsilon=0$) and conceptual representation $Z = \arg\max_yP(Y=y|X)$, we always have $Z=Y$, i.e., representation equals label. Thus we have zero leakage gain $I_{\infty}(X, Z|Y) = 0$ by definition. This contradicts the non-zero leakage gain $I_{\infty}(X, Z|Y) = 1$ under $\varepsilon = 0$ proved in the authors' response. Also, the authors did not answer the question of "what if multiple values of X map to the same representation value".
> > > >
> > > > In general my major concern remains regarding the meaningfulness of the leakage gain metric and resulting tradeoffs. Therefore I will keep the rating.

---

### Author Response · Authors · 2023-11-15

We thank all the reviewers for their time and effort. We are glad that the reviews highlight that understanding of the trade-off between utility and the least-privilege learning principle is an important problem (KceC, qV6w), and that our characterization of this inherent trade-off is interesting (orQ6), insightful (3RFy), and reveals a general property of machine learning (qV6w). The reviews also appreciated that our paper clearly states the worst-case assumptions (3RFy), provides experimental support for the theoretical results (orQ6), and pointed out that the paper is well-written and easy to follow (3RFy, KceC). We respond to the comments and questions to each individual review.

---

### Meta-Review · Area_Chair_Wqx6 · 2023-12-08

**Metareview:**

This paper gives the first theoretical analysis of the least-privilege principle in split learning. The authors prove that the utility of learned representations for a certain task has an inherent trade-off with the amount of information that it reveals about other tasks. The authors also validate their findings empirically on the LFWA+ dataset.

Reviews are split for this paper, with the two most major weaknesses being:
1. The assumptions are quite restrictive, and the authors did not introduce any new analysis technique beyond the standard arguments in information theory.
2. Empirical evaluation is done using an unrealistic setup where the model is split at the last layer. This setup defeats the purpose of split learning since almost all of the model's inference computation is done on the client side. Evaluation is also done using a single model and dataset.

While these two weaknesses are not critical by themselves, their combination reduces the significance of the paper's results. AC believes the paper is not ready for publication at this time, but encourages the authors to revise the draft accordingly to address these weaknesses.

**Justification For Why Not Higher Score:**

The two weaknesses listed above weaken paper's significance. There is no reviewer that strongly supports the paper's acceptance.

**Justification For Why Not Lower Score:**

N/A

---

### Decision · Program_Chairs · 2024-01-16

Reject